# Federated Residual Low-Rank Adaptation of Large Language Models

**Yunlu Yan**[1*]    **Chun-Mei Feng**[2*]    **Wangmeng Zuo** [3,4]    **Rick Siow Mong Goh**[2]
**Yong Liu**[2]    **Lei Zhu**[1,5†]
[1] The Hong Kong University of Science and Technology (Guangzhou), China
[2] Institute of High Performance Computing (IHPC),
Agency for Science, Technology and Research (A*STAR), Singapore
[3] Harbin Institute of Technology, China
[4] Pengcheng Laboratory, China
[5] The Hong Kong University of Science and Technology, China

## Abstract

Low-Rank Adaptation (LoRA) presents an effective solution for federated fine-tuning of Large Language Models (LLMs), as it substantially reduces communication overhead. However, a straightforward combination of FedAvg and LoRA results in suboptimal performance, especially under data heterogeneity. We noted this stems from both intrinsic (*i.e.*, constrained parameter space) and extrinsic (*i.e.*, client drift) limitations, which hinder it effectively learn global knowledge. In this work, we proposed a novel **F**ederated **R**esidual **L**ow-**R**ank **A**daption method, namely FRLoRA, to tackle above two limitations. It directly sums the weight of the global model parameters with a residual low-rank matrix product (*i.e.*, weight change) during the global update step, and synchronizes this update for all local models. By this, FRLoRA performs global updates in a higher-rank parameter space, enabling a better representation of complex knowledge structure. Furthermore, FRLoRA reinitializes the local low-rank matrices with the principal singular values and vectors of the pre-trained weights in each round, to calibrate their inconsistent convergence, thereby mitigating client drift. Our extensive experiments demonstrate that FRLoRA consistently outperforms various state-of-the-art FL methods across nine different benchmarks in natural language understanding and generation under different FL scenarios. Codes are available at https://github.com/IAMJackYan/FRLoRA.

## 1 Introduction

Federated Learning (FL) (Li et al., 2020a; Yin et al., 2021; Li et al., 2021b) allows multiple clients to collaboratively train a globally shared model by transferring model parameters, emerging as an effective distributed solution. However, applying this approach to Large Language Models (LLMs) faces significant challenges due to their enormous parameter sizes (Liu et al., 2021; Ye et al., 2024b). For example, GPT-4 (Achiam et al., 2023) has 1.75 trillion parameters, and even the smallest version of LLaMA-2 (Touvron et al., 2023) has 7 billion parameters. Fully Fine-Tuning (FFT) such large models in a federated setting leads to substantial communication overhead, making it impractical for real-world applications. A promising solution is to integrate Parameter-Efficient Fine-Tuning (PEFT) techniques to reduce the number of trainable parameters in LLMs.

Low-Rank Adaptation (LoRA) (Hu et al., 2021), a state-of-the-art PEFT technique, freezes the pre-trained weights and injects low-rank matrices into specific layers of the model, offering an excellent solution for federated fine-tuning of LLMs. However, we found that a straightforward combination of FedAvg (McMahan et al., 2017) and LoRA (see Figure 1 (a)) *struggle to effectively learn global knowledge* (see Figure 1 (c)), particularly when the data across clients is a non-Independent and

---

*Equal contribution.
†Lei Zhu (leizhu@ust.hk) is the corresponding author.

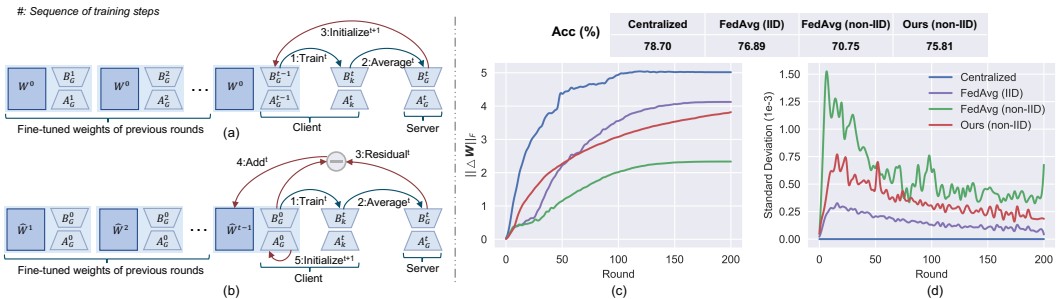

Figure 1: **Illustration of** (a) FedAvg with LoRA and (b) FRLoRA, where superscripts $t$ denote timestamps and $\rightarrow$ highlights their differences. Specifically, FedAvg iteratively updates local and global low-rank matrices, whereas FRLoRA directly updates the parameters of the local and global models $\hat{W}^{t-1}$ by summing their weights with a residual low-rank matrix product. (c) Average Frobenius norm of global $\Delta W$ and (d) Average standard deviation of local low-rank matrices versus the communication rounds, revealing that FRLoRA can effectively learn global knowledge and mitigate client drift. More details can be seen in Appendix §A.

Identically Distribution (non-IID[1]), leading to serious performance degradation. This is contrary to the goal of FL, which aims to represent the learned knowledge of all clients through the global model. The above issue arises from both intrinsic and extrinsic factors. ***Extrinsically***, the data heterogeneity can result in client drift (Karimireddy et al., 2020) (see Figure 1 (d)), and this phenomenon is more pronounced with LoRA compared to FFT, as the parameters in LoRA are locally quadratic (Liu et al., 2024). ***Intrinsically***, both the local and global low-rank matrices are constrained to updates in the parameter space with a rank no greater than $r$. Such constrained parameter space makes it difficult to effectively capture the diverse knowledge of all clients.

While the non-IID has been widely explored in FFT-based FL (Ye et al., 2023), little is understood about how to tackle it in LoRA-based FL. Previous empirical studies (Ye et al., 2024a;b) have indicated that simply integrating LoRA with existing heterogeneous FL methods like FedProx (Li et al., 2020b) and SCAFFOLD (Karimireddy et al., 2020) fails to address data heterogeneity. This primarily arises from the different update patterns and learning capabilities between LoRA and FFT (Liu et al., 2024). Consequently, there is a need for a more nuanced approach to improve the performance of LoRA-based FL.

To this end, we propose a novel **F**ederated **R**esidual **Lo**w-**R**ank **A**daptation approach, namely FRLoRA, to handle the data heterogeneity in LoRA-based FL. Instead of only updating local and global low-rank matrices, during the global update step, FRLoRA (see Figure 1 (b)) directly update the weight of the global model[2] parameters by summing its weights with a residual low-rank matrix product (*i.e.*, weight change) and synchronize this update for all local models, which makes the update pattern similar to that of FFT-based FL. By accumulating residuals with increasing rounds, the global updates can be conducted in higher-rank parameter space, enabling a more flexible representation of complex knowledge structures. Moreover, FRLoRA reinitializes the local low-rank matrices with the principal singular values and vectors of the pre-trained weights in each round. This ensures that the local low-rank matrices are consistently updated in the principal singular space of the pre-trained weights, benefiting to alleviate client drift (see Figure 1 (d)). Based on the above two aspects, FRLoRA effectively captures global knowledge (see Figure 1 (c)), thereby significantly improving the performance of the global model.

**Contribution.** In this work, we improve the LoRA-based FL by addressing both intrinsic and extrinsic limitations, yielding a novel FL method, FRLoRA. Different from previous paradigms that only update local and global low-rank matrices, FRLoRA directly update the weight of the local and global model parameters. We conducted extensive experiments on nine benchmarks across different language tasks under various FL scenarios, including IID and non-IID, as well as partial and full participation, adequately demonstrating the superiority of FRLoRA over the state-of-the-art FL methods.

---

[1]We use "data heterogeneity" and "non-IID" interchangeably.

[2]We term the original part of the LLM without injected low-rank matrices as "model".

## 2 RELATED WORK

### 2.1 LARGE LANGUAGE MODELS

Generally, LLMs, also known as foundational language models (Myers et al., 2024) like GPT-3/4 (Brown, 2020; Achiam et al., 2023), LLaMA-2/3 (Touvron et al., 2023; Dubey et al., 2024), and PaLM (Chowdhery et al., 2023) have learned abundant knowledge by being pre-trained on massive public language corpus (Raffel et al., 2020; Gao et al., 2020), achieving tremendous success in various applications. Their versatility in language tasks has attracted researchers to fine-tune them across a broad spectrum of fields, such as healthcare (Singhal et al., 2023; Xie et al., 2024; Wang et al., 2023), to further enhance their capabilities for specialized downstream tasks. Accompanying this trend is the emergence of various language model fine-tuning methods, such as Instruction-Tuning (IT) (Mishra et al., 2022; Zhang et al., 2023b) and Reinforcement Learning from Human Feedback (RLHF) (Bai et al., 2022). However, all of them train the models using centralized data. In real-world applications, data is typically distributed across different users' edge devices, such as mobile phones and laptops. Due to privacy regulations like GDPR (Voigt & Von dem Bussche, 2017), collecting this data can be expensive and even infeasible. In this work, we aim to explore how to utilize FL to fine-tune LLMs in a distributed and effective manner. This facilitates the effective utilization of discrete data from different users.

### 2.2 PARAMETER-EFFICIENT FINE-TUNING

PEFT (Mangrulkar et al., 2022) has emerged as an important technique for reducing training parameters in fine-tuning large pre-trained models (Xu et al., 2023). Instead of fine-tuning all parameters, PEFT introduces only a small number of trainable parameters at specific locations of networks, such as input tokens (Li & Liang, 2021; Lester et al., 2021) and layers (Houlsby et al., 2019), while keeping the rest of parameters frozen. Recently popular LoRA methods (Hu et al., 2021; Liu et al., 2024; Zhang et al., 2023a; Gao et al., 2024; Lin et al., 2024) fine-tune LLMs by injecting two trainable low-rank matrices into the layers of the base model. It not only effectively compresses training costs but also shows outstanding performance across a range of language tasks. Inspired by this, several works (Wu et al., 2024; Vavekanand & Sam, 2024; Ye et al., 2024b;a) introduced LoRA into FL to decrease the communication cost of federated fine-tuning LLMs. However, most of them ignore data heterogeneity in real-world scenarios, which can significantly degrade the performance of LoRA-based FL. This motivates us to improve the performance of LoRA-based FL under data heterogeneity.

### 2.3 FEDERATED LEARNING

**Federated Fine-tune Large Language Models.** FL has become a de facto distributed solution. Since FL can protect user privacy, it has been widely used in privacy-sensitive areas like finance (Long et al., 2020; Chatterjee et al., 2023) and medical (Feng et al., 2022; Yan et al., 2024a; Jiang et al., 2023; Yan et al., 2024b; Feng et al., 2023a; Yan et al., 2023a). The tremendous success of FL has also attracted the attention of researchers to introduce it into LLMs (Qu, 2024; Fan et al., 2023; Han et al., 2024; Zhang et al., 2024). For example, FederatedScope-LLM (Kuang et al., 2024) built a comprehensive package for federated fine-tuning LLMs and explored the federated instruction tuning. OpenFedLLM (Ye et al., 2024b) provided the empirical results for federated instruction tuning and federated value alignment. Besides, Ye et al. (2024a) provided realistic benchmarks and empirical results for federated fine-tuning LLMs. These foundational studies have greatly advanced the development of this field. In contrast to them, this work focuses on a different challenge: how to effectively fine-tune LLMs, especially under data heterogeneity. Similar to us, Sun et al. (2024) proposed FFA-LoRA to improve the performance of LoRA-based FL under differential privacy and data heterogeneity by fixing the matrix $A$ in LoRA. This can further diminish the learning capability of LoRA, resulting in degraded performance under realistic heterogeneous FL settings.

**Data Heterogeneity in Federated Learning.** The standard FL process was introduced by FedAvg (McMahan et al., 2017), a pioneer algorithm, which updates the global model by averaging the weights of local model parameters. Despite its success, the performance of FedAvg is inevitably compromised by practical challenges such as data heterogeneity (Zhao et al., 2018; Ye et al., 2023) or model heterogeneity (Alam et al., 2022). Data heterogeneity (Ye et al., 2023; Feng et al., 2023b) is a

fundamental challenge in FL, with a typical example being the non-IID across different clients (Zhao et al., 2018). This issue causes local model bias due to skewed data, which affects global aggregation and leads to significant degradation in the global model's performance, a phenomenon known as client drift. (Karimireddy et al., 2020; Li et al., 2022). To address this, several studies have proposed enhanced versions of FedAvg, either by modifying local training (Li et al., 2021a; Tan et al., 2022; Luo et al., 2021; Zhang et al., 2022; Gao et al., 2022; Yan et al., 2023b), global aggregation (Li et al., 2020c; Wang et al., 2020; Li et al., 2023; Fallah et al., 2020), or both (Jiang et al., 2022; Huang et al., 2023). However, these enhancements primarily target data heterogeneity in FFT-based FL. Applying them to LoRA-based FL may not yield favourable results due to the distinct update patterns and learning capabilities between LoRA and FFT (Liu et al., 2024). As evidence, previous empirical studies (Ye et al., 2024a;b) have shown that combining several well-known heterogeneous FL algorithms (*e.g.*, FedProx (Li et al., 2020b) and SCAFFOLD (Karimireddy et al., 2020)) with LoRA can result in worse performance than FedAvg+LoRA. Therefore, how to address the data heterogeneity in LoRA-based FL remains an unsolved question, which is the main goal of this work.

## 3 METHODOLOGY

### 3.1 PRELIMINARY

**Low-Rank Adaptation.** When fine-tuning LLMs on downstream tasks, LoRA (Hu et al., 2021) only updates the parameters of some layers, *e.g.*, self-attention. Consider a layer in the network, LoRA freezes the pre-trained weight $\boldsymbol{W}^0 \in \mathbb{R}^{d_1 \times d_2}$ and leans the weight change $\Delta \boldsymbol{W}$ by injecting two trainable low-rank matrices $\boldsymbol{B} \in \mathbb{R}^{d_1 \times r}$ and $\boldsymbol{A} \in \mathbb{R}^{r \times d_2}$, where rank $r \ll \min(d_1, d_2)$. The updated weight is expressed as:

$$\widetilde{\boldsymbol{W}} = \boldsymbol{W}^0 + \Delta \boldsymbol{W} = \boldsymbol{W}^0 + \boldsymbol{B}\boldsymbol{A}. \tag{1}$$

To ensure consistency with the pre-trained weight during the initial phase, $\boldsymbol{B}$ is initialized as a zero matrix, while $\boldsymbol{A}$ is initialized with Gaussian noise $\mathcal{N}(0, \sigma^2)$. Besides, a hyper-parameter $\alpha$ is typically used to scale $\Delta \boldsymbol{W}$, and the scaling factor is $\frac{\alpha}{r}$.

**Federated Learning with Low-Rank Adaptation.** Consider a FL system with $K$ clients and a central server, each client $k \in [K]$ has a private dataset $D_k$ for downstream tasks such as natural language understanding or generation. Each dataset $D_k$ contains $n_k$ training samples $\{(x_i, y_i)\}_{i=1}^{n_k}$ and our goal is to federated fine-tune a global LLM $f(\boldsymbol{W}^0)$ on these discrete data. Due to the large size of LLMs, we cannot train and transmit all model parameters. A straightforward approach is to directly combine FedAvg with LoRA (Ye et al., 2024b;a), where each client maintains local low-rank matrices and optimizes them by minimizing the empirical risk:

$$\mathcal{F}_G = \frac{1}{K} \sum_{k=1}^{n_k} \mathcal{F}_k, \quad \text{and} \ \mathcal{F}_k = \frac{1}{n_k} \sum_{(x_i, y_i) \sim D_k} \ell(f(x_i; \boldsymbol{W}^0; \boldsymbol{B}_k^t; \boldsymbol{A}_k^t); y_i), \tag{2}$$

where $\mathcal{F}_G$ and $\mathcal{F}_k$ are global and local objectives, $\boldsymbol{B}_k^t$ and $\boldsymbol{A}_k^t$ are the low-rank matrices at $t$-th round training, and $\ell$ is the loss function. All updated local low-rank matrices will be transferred to the server and we can get the global low-rank matrices:

$$\boldsymbol{B}_G^t = \frac{1}{K} \sum_{k=1}^{K} \boldsymbol{B}_k^t, \quad \boldsymbol{A}_G^t = \frac{1}{K} \sum_{k=1}^{K} \boldsymbol{A}_k^t. \tag{3}$$

The updated global low-rank matrices $\boldsymbol{B}_G^t$ and $\boldsymbol{A}_G^t$ will be returned to each client as the initialization for the next round of training, where $\boldsymbol{B}_G^0 \sim 0$ and $\boldsymbol{A}_G^0 \sim \mathcal{N}(0, \sigma^2)$. The weight of final fine-tuned global model parameters after $T$ rounds can be written as:

$$\widetilde{\boldsymbol{W}}^T = \boldsymbol{W}^0 + \Delta \boldsymbol{W}^T = \boldsymbol{W}^0 + \boldsymbol{B}_G^T \boldsymbol{A}_G^T. \tag{4}$$

**Limitation.** The above method is easy to implement and can significantly reduce communication overhead when fine-tuning LLMs. However, it fails to effectively capture global knowledge, especially under data heterogeneity. This issue arises from two factors: ❶ **constrained parameter space** and ❷ **client drift**, where ❶ is the intrinsic limitation of LoRA-based FL and ❷ is the extrinsic influence caused by data heterogeneity.

---

**Algorithm 1:** FRLoRA

---

**Input:** Number of clients $K$, communication rounds $T$, learning rate $\eta$, Pre-trained weight $\boldsymbol{W}^0$, Datasets $D_1, D_2, \ldots, D_K$, rank $r$

**Output:** Fine-tuned weight $\widetilde{\boldsymbol{W}}^T$

1 **Server-side Execution:**
2 $\boldsymbol{USV} \leftarrow SVD(\boldsymbol{W}^0)$
3 $\boldsymbol{B}_G^0 \leftarrow \boldsymbol{U}[:r]\sqrt{\boldsymbol{S}[:r]}, \quad \boldsymbol{A}_G^0 \leftarrow \sqrt{\boldsymbol{S}[:r]}\boldsymbol{V}[:r]$ // Global initialize for LoRA
4 $\hat{\boldsymbol{W}}^0 \leftarrow \boldsymbol{W}^0 - \boldsymbol{B}_G^0\boldsymbol{A}_G^0$ // Consistent with pre-trained model
5 **for** *round $t = 1, 2, ..., T$* **do**
6      **for** *client $k = 1, 2, ..., K$ **parallelly*** **do**
7         $\boldsymbol{B}_k^t, \boldsymbol{A}_k^t \leftarrow$ **Local Training** $(k, \boldsymbol{B}_G^{t-1}, \boldsymbol{A}_G^{t-1}, t)$
8      **end**
9      $\boldsymbol{B}_G^t \leftarrow \frac{1}{K}\sum_{k=1}^K \boldsymbol{B}_k^t, \quad \boldsymbol{A}_G^t \leftarrow \frac{1}{K}\sum_{k=1}^K \boldsymbol{A}_k^t$ // Parameter aggregation
10      $\Delta\boldsymbol{W}^t \leftarrow \boldsymbol{B}_G^t\boldsymbol{A}_G^t - \boldsymbol{B}_G^0\boldsymbol{A}_G^0, \quad \hat{\boldsymbol{W}}^t \leftarrow \hat{\boldsymbol{W}}^{t-1} + \Delta\boldsymbol{W}^t$ // $\hat{\boldsymbol{W}}^t, \Delta\boldsymbol{W}^t$ keep consistent between server and clients, we thus use same symbol
11 **end**
12 $\widetilde{\boldsymbol{W}}^T \leftarrow \hat{\boldsymbol{W}}^T + \boldsymbol{B}_G^0\boldsymbol{A}_G^0$
13 return $\widetilde{\boldsymbol{W}}^T$ // Fine-tuned weight
14 **Local Training** $(k, \boldsymbol{B}_G^{t-1}, \boldsymbol{A}_G^{t-1}, t)$:
15 Save $\boldsymbol{B}_G^0$ and $\boldsymbol{A}_G^0$
16 $\boldsymbol{B}_k^t \leftarrow \boldsymbol{B}_G^0, \boldsymbol{A}_k^t \leftarrow \boldsymbol{A}_G^0$ // Local initialize for LoRA
17 **for** $(x_i, y_i) \sim D_k$ **do**
18      $\mathcal{F}_k \leftarrow \ell(f(x_i; \hat{\boldsymbol{W}}^{t-1}; \boldsymbol{B}_k^t; \boldsymbol{A}_k^t); y_i)$
19      $\boldsymbol{B}_k^t\boldsymbol{A}_k^t \leftarrow \boldsymbol{B}_k^t\boldsymbol{A}_k^t - \eta\nabla\mathcal{F}_k$
20 **end**
21 return $\boldsymbol{B}_k^t$ and $\boldsymbol{A}_k^t$ to server

---

## 3.2 Federated Residual Low-rank Adaptation

To tackle the above two limitations in LoRA-based FL, we propose a new FL method, FRLoRA. Algorithm 1 shows its whole training procedure, and we describe each part of our framework in detail below.

**Initialization in Principal Singular Space.** LoRA typically initializes $\boldsymbol{B}$ as 0 and $\boldsymbol{A}$ with Gaussian noise. Such initialization makes convergence challenging, and optimizing with the data heterogeneity results in inconsistent convergence rates across different clients, further exacerbating client drift. This hinders the global $\Delta\boldsymbol{W}$ from capturing task-specific information. To this end, we initialize $\boldsymbol{B}$ and $\boldsymbol{A}$ in principal singular space of the pre-trained weight inspired by Meng et al. (2024), enabling faster learning of task-specific information for downstream tasks. Specifically, we first decompose the pre-trained weights using Singular Value Decomposition (SVD) as follows:

$$\boldsymbol{USV} = SVD(\boldsymbol{W}^0), \tag{5}$$

where $\boldsymbol{U} \in \mathbb{R}^{d_1 \times d_1}$ and $\boldsymbol{V} \in \mathbb{R}^{d_2 \times d_2}$ are the singular vectors with orthonormal columns, and $\boldsymbol{S} \in \mathbb{R}^{d_1 \times d_2}$ is a diagonal matrix containing the singular values in descending order. We then use the principal singular values and vectors to initialize $\boldsymbol{B}_G^0$ and $\boldsymbol{A}_G^0$, which can be written as:

$$\boldsymbol{B}_G^0 = \boldsymbol{U}[:r]\sqrt{\boldsymbol{S}[:r]}, \quad \boldsymbol{A}_G^0 = \sqrt{\boldsymbol{S}[:r]}\boldsymbol{V}[:r]. \tag{6}$$

$\boldsymbol{B}_G^0$ and $\boldsymbol{A}_G^0$ will be sent to each client to initialize the local low-rank matrices. Additionally, to ensure consistency, the weight of the global model parameters will be updated accordingly:

$$\hat{\boldsymbol{W}}^0 = \boldsymbol{W}_0 - \boldsymbol{B}_G^0\boldsymbol{A}_G^0. \tag{7}$$

**Local Update Step.** Each client performs local training on its dataset and the local objective of Eq. (2) can be rewritten as:

$$\mathcal{F}_k = \frac{1}{n_k} \sum_{(x_i, y_i) \sim D_k} \ell(f(x_i; \hat{\boldsymbol{W}}^{t-1}; \boldsymbol{B}_k^t; \boldsymbol{A}_k^t); y_i). \tag{8}$$

We send the optimized local low-rank matrices $\boldsymbol{B}_k^t$ and $\boldsymbol{A}_k^t$ to the server, which then uses Eq. (3) to update the global low-rank matrices.

**Global Update Step.** Different from the traditional LoRA-based FL methods, FRLoRA directly updates the weight of the global model parameters by adding a residual of the low-rank matrix product, which bridges the gap with FFT-based FL.

After getting global low-rank matrices $\boldsymbol{B}_G^t$ and $\boldsymbol{A}_G^t$, we can get a residual of the low-rank matrix product as:

$$\Delta \boldsymbol{W}^t = \boldsymbol{B}_G^t \boldsymbol{A}_G^t - \boldsymbol{B}_G^0 \boldsymbol{A}_G^0. \tag{9}$$

The residual matrix $\Delta \boldsymbol{W}^t$ represents the weight change learned in this round, and it will be added to the weights of the global model parameters:

$$\hat{\boldsymbol{W}}^t = \hat{\boldsymbol{W}}^{t-1} + \Delta \boldsymbol{W}^t. \tag{10}$$

The updated weight $\hat{\boldsymbol{W}}^t$ of the global model parameters will be frozen again, and the local low-rank matrices of FRLoRA will be reinitialized to $\boldsymbol{B}_G^0$ and $\boldsymbol{A}_G^0$, preparing for the next round of training. This makes the update pattern of FRLoRA similar to that of FFT-based FL, where the weight of model parameters from the previous round serves as the starting point for the next round of training.

**Parameter Synchronization.** Notably, the process of Eq. (7) and Eq. (10) are synchronized across clients. As a result, each local model remains consistent with the global model, and we use the same symbol $\hat{\boldsymbol{W}}^t$ to represent them. The final fine-tuned weight of Eq. (4) can be rewritten as:

$$\widetilde{\boldsymbol{W}}^T = \hat{\boldsymbol{W}}^T + \boldsymbol{B}_G^0 \boldsymbol{A}_G^0. \tag{11}$$

Analysis in §3.3 reveals that FRLoRA enables the global model to be updated in a higher-rank space, effectively addressing ***Limitation ❶***. Besides, reinitialization can be viewed as a recalibration of local and global low-rank matrices, ensuring that the local training of each client always occurs in the same global principal singular space, which benefits mitigate client drift (***Limitation ❷***).

## 3.3 THEORETICAL INSIGHTS

In this section, we conduct a theoretical analysis of the update patterns of FRLoRA and LoRA-based FL. For LoRA-based FL, the final weight change of the global model parameters is represented by $\boldsymbol{B}_G^T \boldsymbol{A}_G^T$. Applying the basic rank inequality, we obtain:

$$\text{rank}(\boldsymbol{B}_G^T \boldsymbol{A}_G^T) \leq \min(\text{rank}(\boldsymbol{A}_G^T), \text{rank}(\boldsymbol{B}_G^T)), \tag{12}$$

where $\text{rank}(\boldsymbol{A}_G^T) \leq \min(r, d_2)$ and $\text{rank}(\boldsymbol{B}_G^T) \leq \min(d_1, r)$. Consequently, since LoRA-based FL only updates the low-rank matrices, the global updates are restricted to a low-rank subspace with a rank no greater than $r$.

In contrast, the final fine-tuned weight of FRLoRA can be expressed as:

$$\widetilde{\boldsymbol{W}}^T = \boldsymbol{W}^0 + \underbrace{\Delta \boldsymbol{W}^1 + \Delta \boldsymbol{W}^2 + \ldots + \Delta \boldsymbol{W}^T}_{\text{residual accumulation}}. \tag{13}$$

We also present the following additive rank inequality:

**Lemma 1.** *(Additive Rank Inequality) For any two matrices $\boldsymbol{M}_1$ and $\boldsymbol{M}_2$, it holds that:*

$$rank(\boldsymbol{M}_1 + \boldsymbol{M}_2) \leq rank(\boldsymbol{M}_1) + rank(\boldsymbol{M}_2). \tag{14}$$

Given that FRLoRA reinitializes in each training round and that the local and global updates for each round are independent, the rank of the residual accumulation increases with the number of rounds according to Lemma 1. This implies that global updates occur in a higher-rank parameter space, which enables the global model to capture more complex knowledge structures.

From another perspective, $\Delta \boldsymbol{W}^t$ can be interpreted as the total gradient of the global model over a round by minimizing $\mathcal{F}_G$. As FRLoRA learns in the principal singular space of the pre-trained model during each round, the gradient $\Delta \boldsymbol{W}^t$ of FRLoRA is larger compared to that of LoRA-based FL, based on the analysis of Meng et al. (2024). This allows the global model to acquire more task-specific knowledge, as illustrated in Figure 1 (c).

# 4 EXPERIMENTS

To validate the effectiveness of our method. We conducted comprehensive experiments across both Natural Language Understanding (NLU) and Natural Language Generation (NLG) tasks. We compared our method against 9 state-of-the-art baselines, including **FedAvg** (McMahan et al., 2017), **FedProx** (Li et al., 2020b), **SCAFFOLD** (Karimireddy et al., 2020), **FedAvgM** (Hsu et al., 2019), **FedAdagard** (Reddi et al., 2021), **FedYogi** (Reddi et al., 2021), **FedAdam** (Reddi et al., 2021), **FlexLoRA** (Bai et al., 2024) and **FFA-LoRA** (Sun et al., 2024). The experiments involved 4 NLU benchmarks: **RTE** (Wang et al., 2019), **COLA** (Wang et al., 2019), **20NG** (Lang, 1995) and **QNLI** (Wang et al., 2019), as well as 5 NLG benchmarks: **MetaMathQA** (Yu et al., 2023), **Alpaca-GPT4** (Peng et al., 2023), **FedAya** (Ye et al., 2024a), **Fed-ChatbotIT** (Ye et al., 2024a), and **Fed-WildChat** (Ye et al., 2024a). All experiments were implemented using PyTorch and conducted on an NVIDIA A100 GPU with 40 GB of memory. Due to page limitations, more details and results are presented in the *Appendix* §B and §C.

## 4.1 QUANTITATIVE COMPARISON ON NLU TASKS

**Task Setup.** Following Kuang et al. (2024), we randomly partition the training set of each benchmark using the Dirichlet distribution sampling ($D_k \sim Dir(\beta)$), which is a commonly employed strategy for simulating realistic data heterogeneity (Ye et al., 2023). The level of data heterogeneity is controlled by $\beta$, where a smaller $\beta$ means higher heterogeneity. In our experiments, $\beta$ is set to 0.5. We simulate a scenario with 5 clients, all of which participate in training during each round. To evaluate the performance of all methods, we use the validation sets of RTE, COLA, and QNLI, and the test set of 20NG. Accuracy is used as the evaluation metric for RTE, QNLI, and 20NG, while the Matthews correlation coefficient is employed for COLA.

**Implementation Details.** We utilize RoBERTa-base (Liu, 2019) for the NLU tasks. For LoRA, we set the parameter $r$ to 16 and $\alpha$ to 32. The AdamW optimizer is used with a batch size of 64, a learning rate of 2e-4 and cosine annealing schedules. All methods are trained for 200 rounds. The local update step is set to 10 for RTE and 30 for QNLI, 20NG, and COLA based on the quantity of data in each dataset.

Table 1: **Experimental results on varios NLU benchmarks**. Best results are marked in **bold**.

| Method | RTE | COLA | 20NG | QNLI |
|---|---|---|---|---|
| FedAvg | 70.75 | 63.96 | 66.83 | 90.37 |
| FedProx | 69.31 | 63.41 | 66.48 | 90.05 |
| SCAFFOLD | 67.15 | 62.83 | 65.78 | 89.60 |
| FedAvgM | 61.73 | 43.82 | 67.04 | 89.91 |
| Fedadagrad | 66.42 | 54.27 | 63.48 | 87.57 |
| FedYogi | 63.53 | 55.20 | 66.44 | 86.21 |
| FedAdam | 66.06 | 56.30 | 66.94 | 89.69 |
| FFA-LoRA | 68.69 | 58.43 | 66.88 | 89.33 |
| FlexLoRA | 70.28 | 62.56 | 65.98 | 90.03 |
| **FRLoRA (Ours)** | **75.81** | **64.80** | **69.41** | **91.10** |

**Results.** As shown in Table 1, directly integrating heterogeneous FL methods with LoRA does not achieve better performance on the NLU task compared to FedAvg. This is primarily due to the differing update pattern between FFT and LoRA, which is consistent with previous empirical findings (Ye et al., 2024b;a). Furthermore, we observe that FFA-LoRA performs worse than FedAvg in our experimental setting. This is due to FFA-LoRA's focus on differential privacy, as well as the discrepancy between our task setup and its original configuration, which employed manually partitioned data. Such partitioning does not fully reflect the complexity of real-world scenarios, leading to a mismatch in performance under our experimental settings. The results in the real-world data heterogeneity setting suggest that updating only matrix $B$ in FFA-LoRA fails to effectively capture global knowledge. In contrast, our meticulous design enables FRLoRA to consistently outperform all baselines. It achieves significant improvements over FedAvg, such as increasing performance from **70.35**% to **75.81**% on RTE and from **66.83**% to **69.41**% on 20NG. This demonstrates that FRLoRA can effectively address the issue of data heterogeneity in NLU tasks.

Table 2: **Experimental results on NLG tasks**, where **MetaMathQA** and **Alpaca-GPT4** are used as the training dataset, respectively. Results are copied from (Ye et al., 2024b). **Avg.** is the average result of corresponding metrics, and the best results are marked in **bold**.

| Method | MetaMathQA | | | Alpaca-GPT4 | | | | |
|---|---|---|---|---|---|---|---|---|
| | GSM8K | Math | Avg. | Vicuna | MT-1 | MT-2 | MT-Avg | Avg. |
| FedAvg | 34.95 | 4.48 | 19.71 | 7.925 | 4.650 | 2.025 | 3.346 | 4.486 |
| FedProx | 35.40 | 4.66 | 20.03 | 7.875 | 4.538 | 1.848 | 3.201 | 4.320 |
| SCAFFOLD | 35.78 | 5.08 | 20.43 | 7.675 | 4.689 | 2.288 | 3.488 | 4.535 |
| FedAvgM | 34.79 | 4.64 | 19.71 | 7.938 | **4.838** | 2.038 | 3.456 | 4.567 |
| FedAdagrad | 29.64 | 4.06 | 16.85 | 7.931 | 4.675 | 2.025 | 3.350 | 4.495 |
| FedYogi | 30.09 | 4.04 | 17.06 | 8.031 | 4.550 | 1.938 | 3.244 | 4.440 |
| FedAdam | 31.84 | 4.12 | 17.98 | 7.975 | 4.650 | 2.175 | 3.413 | 4.508 |
| FFA-LoRA | 28.05 | 3.78 | 15.91 | 7.862 | 4.712 | 1.950 | 3.331 | 4.463 |
| FlexLoRA | 34.09 | 4.31 | 19.20 | 7.884 | 4.561 | 2.012 | 3.286 | 4.435 |
| **FRLoRA (Ours)** | **44.27** | **5.22** | **24.74** | **8.044** | 4.775 | **2.481** | **3.635** | **4.733** |

Table 3: **Experimental results on NLG tasks**, where **Fed-Aya** is used as the training dataset and evaluated on **Ref-GPT4**. Results are copied from (Ye et al., 2024a). **Avg.** is the average result of all evaluation metrics, and the best results are marked in **bold**.

| Method | ar | en | es | fr | pt | ru | te | zh | Avg. |
|---|---|---|---|---|---|---|---|---|---|
| FedAvg | 2.50 | 8.00 | 5.50 | 5.35 | 4.95 | 5.65 | **2.00** | 5.25 | 4.90 |
| FedProx | 3.20 | 7.10 | 5.90 | 5.65 | 4.85 | 5.20 | 1.60 | 5.80 | 4.92 |
| SCAFFOLD | 2.65 | 7.75 | **6.30** | 5.35 | 5.00 | **6.35** | 1.45 | 4.90 | 4.97 |
| FedAvgM | 3.00 | 7.80 | 5.35 | 5.00 | **5.30** | 5.65 | 1.90 | 5.00 | 4.86 |
| FedAdagrad | 2.50 | 7.85 | 5.15 | 5.25 | 4.45 | 5.75 | 1.55 | 5.50 | 4.75 |
| FedYogi | 2.00 | 8.45 | 6.15 | 4.55 | 3.85 | 6.30 | 1.65 | 4.93 | 4.73 |
| FedAdam | 2.40 | **8.50** | 5.25 | 4.70 | 4.35 | 5.40 | 1.90 | 5.20 | 4.71 |
| FFA-LoRA | 2.10 | 7.90 | 5.70 | 5.15 | 4.30 | 5.15 | 1.65 | 4.60 | 4.56 |
| FlexLoRA | 2.60 | 8.20 | 5.25 | 5.05 | 4.70 | 5.20 | 1.85 | 4.75 | 4.70 |
| **FRLoRA (Ours)** | **4.45** | 7.75 | 6.15 | **6.65** | 4.75 | 6.25 | 1.55 | **6.95** | **5.56** |

## 4.2 QUANTITATIVE COMPARISON ON NLG TASKS

**Task Setup.** Our experiments for NLG tasks follow previous empirical studies (Ye et al., 2024a;b). For MetaMathQA and Alpaca-GPT4, we partition the datasets in an IID manner, with 10 clients for MetaMathQA and 20 clients for Alpaca-GPT4. In each round, we randomly select 2 clients to participate in training. Fed-Aya, Fed-ChatbotIT and Fed-WildChat are realistic benchmarks with data heterogeneity, consisting of 38, 237, and 100 clients, respectively. Correspondingly, we randomly select 4, 10, and 5 clients to participate in training each round.

**Evaluation.** The evaluation process of NLG tasks is fundamentally different from NLU tasks. We use the following benchmarks for evaluation: **GSM8K** (Cobbe et al., 2021) **Math** (Yu et al., 2024), **Ref-GPT4** (Ye et al., 2024a), **MT-Bench** (Zheng et al., 2024) and **Vicuna** (Chiang et al., 2023). For GSM8K and Math, we use accuracy as the evaluation metric. For the remaining benchmarks, we employ GPT-4 (Achiam et al., 2023) to rate the generated responses on a scale from 1 to 10. All benchmarks are open-ended evaluations. MT-Bench assesses both one-turn and two-turn conversations, with MT-1 and MT-2 representing the scores for one-turn and two-turn interactions, respectively. Other benchmarks focus solely on one-turn conversations.

**Implementation Details.** We employ LLaMA-2-7B (Touvron et al., 2023) for NLG tasks. For MetaMathQA and Alpaca-GPT4, we set $r$ to 32 and $\alpha$ to 64. For the remaining three benchmarks, $r$

Table 4: **Experimental results on NLG tasks**, where **Fed-ChatbotIT** and **Fed-WildChat** are used as training dataset, respectively. Results are copied from (Ye et al., 2024a). **Avg.** is the average result of corresponding metrics, and the best results are marked in **bold**.

| Method | Fed-ChatbotIT | | | | Fed-WildChat | | | |
|---|---|---|---|---|---|---|---|---|
| | MT-1 | Vicuna | Ref-GPT4 | Avg. | MT-1 | Vicuna | Ref-GPT4 | Avg. |
| FedAvg | 4.30 | 6.93 | 5.29 | 5.51 | 4.81 | 7.99 | 5.88 | 6.22 |
| FedProx | 4.25 | 7.21 | 5.00 | 5.49 | 4.86 | 7.93 | 5.74 | 6.17 |
| SCAFFOLD | 3.86 | 7.35 | 4.82 | 5.34 | 4.78 | 7.93 | 5.57 | 6.09 |
| FedAvgM | 4.34 | 7.17 | 4.76 | 5.42 | 4.52 | 8.07 | 5.85 | 6.14 |
| FedAdagrad | 3.94 | **7.50** | 4.99 | 5.48 | 4.76 | 7.76 | 5.93 | 6.14 |
| FedYogi | 4.13 | 7.20 | 5.00 | 5.44 | 4.78 | 8.04 | 5.48 | 6.10 |
| FedAdam | 3.88 | 7.32 | 5.02 | 5.41 | 4.54 | 8.03 | 5.68 | 6.08 |
| FFA-LoRA | 3.78 | 6.85 | 5.45 | 5.36 | 4.37 | 7.79 | 5.57 | 5.91 |
| FlexLoRA | 4.17 | 7.02 | 5.40 | 5.53 | 4.88 | 7.91 | 5.78 | 6.19 |
| **FRLoRA (Ours)** | 4.31 | 7.49 | **5.62** | **5.80** | 4.64 | **8.24** | **7.00** | **6.63** |

Figure 2: **Illustration of global training loss** versus communication rounds on (a) **RTE**, (b) **20NG**, and (c) **MetaMathQA**.

and $\alpha$ are set to 16 and 32, respectively. The AdamW optimizer is used with a learning rate of 5e-4 for MetaMathQA and Alpaca-GPT4, and 2e-4 for other benchmarks, following a cosine annealing schedule. We conduct training with rounds of either 100 or 200. For all benchmarks, IT is employed as the fine-tuning paradigm, utilizing the instruction template of Alpaca (Taori et al., 2023).

**Results.** Table 2 depicts the results of all methods on MetaMathQA and Alpaca-GPT4 benchmarks with IID distribution. In this setting, the impact of client drift is decreased. However, LoRA-based FL still suffers challenges from the intrinsic limitation. In contrast, FRLoRA can effectively address it, resulting in significant performance improvements over all baselines. For example, it improves the average accuracy from **19.71**% to **24.74**% on MetaMathQA. Furthermore, Tables 3 and 4 show the performance of all methods on three realistic benchmarks, *i.e.*, Fed-Aya, FedChatbotIT and Fed-WildChat, with data heterogeneity. Although FRLoRA performs lower than the best baselines on some metrics, it consistently outperforms all baselines on the average metrics across various benchmarks. The above results show the superiority of FRLoRA compared to all baselines, demonstrating its effectiveness in addressing data heterogeneity in NLG tasks.

Moreover, the experiments are conducted in the partial participation setting, demonstrating that FRLoRA can effectively learn global knowledge even when only a subset of clients participates in training. This expands the range of scenarios where our method can be applied. In contrast, FFA-LoRA yields worse performance on five benchmarks compared to FedAvg, as freezing matrix $A$ restricts its learning capability, particularly in the partial participation setting.

### 4.3 CONVERGENCE

In Figure 2, we show the global training loss, *i.e.*, $\mathcal{F}_G$ in Eq. (2), of all FL methods versus communication rounds on RTE, 20NG, and MetaMathQA. Since FRLoRA train low-rank matrices in

Table 5: **Results of ablation study** on **RTE**, **20NG**, **MetaMathQA**, and **Fed-WildChat**. Best results are marked in **bold**.

| Method | NLU | | MetaMathQA | | | Fed-WildChat | | | |
|---|---|---|---|---|---|---|---|---|---|
| | RTE | 20NG | GSM8K | Math | Avg. | MT-1 | Vicuna | Ref-GPT4 | Avg. |
| FedAvg | 70.75 | 66.83 | 34.95 | 4.48 | 19.71 | 4.81 | 7.99 | 5.88 | 6.22 |
| FRLoRA-v1 | 74.12 | 67.68 | 36.02 | 4.73 | 20.37 | 4.84 | 7.96 | 5.91 | 6.23 |
| FRLoRA-v2 | 69.81 | 65.77 | 36.81 | 4.92 | 20.86 | 4.58 | 7.84 | 6.08 | 6.16 |
| FRLoRA-v3 | 58.62 | 63.19 | 39.08 | 5.01 | 22.04 | 4.43 | 8.03 | 6.34 | 6.26 |
| **FRLoRA (Ours)** | **75.81** | **69.41** | **44.27** | **5.22** | **24.74** | 4.64 | **8.24** | **7.00** | **6.63** |

the principal singular space, it exhibits a faster convergence rate compared to other methods. This further presents the superiority of our method. Moreover, as illustrated in Figure 2 (a), the learning mechanism of FRLoRA leads to more stable convergence.

## 4.4 Ablation Study

**Effectiveness of Update Mechanism.** To access the effectiveness of update mechanism in FRLoRA, we built up FRLoRA-v1: FedAvg initializes $B_k^0$ and $A_k^0$ using the principal singular values and vectors of pre-trained weight, *i.e.*, FedAvg+PiSSA (Meng et al., 2024). The results in Table 5 show that FRLoRA yields consistent improvements over FRLoRA-v1 on all datasets across different tasks. This strongly demonstrates the effectiveness of update mechanism in FRLoRA, which enables the global model to be updated in higher-rank parameter space, allowing a better capture of the diverse knowledge from all clients.

**Effectiveness of Initialization Mechanism.** To further explore the effectiveness of initialization mechanism in FRLoRA, we constructed FRLoRA-v2, a variant that employs standard initialization methods, *i.e.*, zero and Gaussian noise, to reinitialize $B_k^t$ and $A_k^t$ at each round. Additionally, we developed FRLoRA-v3, a variant that the weight of local and global model parameters is updated directly using the global low-rank matrix and the Eq. (10) can be rewritten as $\hat{W}^t = \hat{W}^{t-1} + B_G^t A_G^t$. Following, we reinitialize $B_k^{t+1}$ and $A_k^{t+1}$ with the principal singular values and vectors of $W^t$ similar to Eq. (5)-(7).

As shown in Table 5, the performance of FRLoRA-v2 suggests that relying solely on the parameter residual mechanism is insufficient to handle data heterogeneity, as standard initialization struggles to converge properly, resulting in the degradation of global knowledge $\Delta W^t$. As an excellent complement, reinitializing in the principal singular space at each round can effectively address this issue and significantly improve the performance under both IID and non-IID settings. Additionally, a comparison between FRLoRA and FRLoRA-v3 reveals that using SVD to decompose new model weights for initialization at each round results in worse performance compared to our approach. This is attributed to the instability of the SVD operation, using frequently can significantly affect model convergence. Additionally, it also incurs a large amount of computational overhead.

## 5 Conclusion

In this work, we address the challenge of data heterogeneity in LoRA-based FL. We observe that LoRA struggles to effectively capture global knowledge in heterogeneous FL settings due to both intrinsic and extrinsic limitations. To overcome these limitations, we introduce a novel approach, FRLoRA, which directly updates model parameters by learning a residual low-rank matrix product in the principal singular space of pre-trained weight. We validate the effectiveness of FRLoRA through extensive experiments across nine benchmarks from different language tasks. Furthermore, we offer a deeper understanding of our method from both theoretical and empirical perspectives.

## ACKNOWLEDGEMENTS

This work is supported by the Major Key Project of PCL under Grant PCL2024A06, the National Research Foundation, Singapore under its AI Singapore Programme (AISG Award No: AISG2-TC-2021-003), Agency for Science, Technology and Research (A*STAR) through its AME Programmatic Funding Scheme Under Project A20H4b0141, A*STAR Central Research Fund "A Secure and Privacy Preserving AI Platform for Digital Health", and Agency for Science, Technology and Research (A*STAR) through its RIE2020 Health and Biomedical Sciences (HBMS) Industry Alignment Fund Pre-Positioning (IAF-PP) (grant no. H20C6a0032), National Natural Science Foundation of China (Project No. 61902275), Guangdong Provincial Key Lab of Integrated Communication, Sensing and Computation for Ubiquitous Internet of Things(No.2023B1212010007), and the Guangdong Science and Technology Department (No. 2024ZDZX2004).

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

Table 6: **Detailed information of NLU datasets.**

| Datasets | RTE | COLA | QNLI | 20NG |
|---|---|---|---|---|
| Task | Inference | Linguistic acceptability | Inference | Text classification |
| # Labels | 2 | 2 | 2 | 20 |
| # Train | 2.49k | 8.55k | 105k | 11.31k |
| # Test | 277 | 1.04k | 5.46k | 7.53k |
| # Clients (Total) | 5 | 5 | 5 | 5 |
| # Clients (Train) | 5 | 5 | 5 | 5 |
| Metric | Accuracy | Matthews correlation | Accuracy | Accuracy |

Table 7: **Detailed information of NLG datasets.**

| Datasets | MetaMathQA | Alpaca-GPT4 | Fed-Aya | Fed-ChatbotIT | Fed-WildChat |
|---|---|---|---|---|---|
| Task | Question answering | Single-turn chat | Single-turn chat | Single-turn chat | Multi-turn chat |
| Domain | Math | General | Multilingual | General | Multilingual |
| # Train | 395k | 52k | 25,513 | 6166 | 52,703 |
| # Clients (Total) | 10 | 20 | 38 | 237 | 100 |
| # Clients (Train) | 2 | 2 | 4 | 10 | 5 |

## A   EMPIRICAL ANALYSIS

In this section, we conduct an empirical analysis to reveal the update pattern of LoRA in FL. Drawing on RTE as the running example, we first record the average Frobenius norm of global $\Delta W$, which serves as an indicator of the task-specific knowledge that LoRA acquires for downstream tasks (Si et al., 2024). Notably, the global $\Delta W$ for FRLoRA is the residual accumulation term in Eq. (13). Additionally, we track the standard deviation among all local low-rank matrices across clients during the update process, which provides a measure of the level of client drift.

The results versus the communication rounds, illustrated in Figure 1 (c) and (d), show that data heterogeneity hinders LoRA from effectively capturing global knowledge and incurs significant client drift among local low-rank matrices, resulting in serious performance degradation from **76.89**% to **70.75**%. In contrast, our proposed method, FRLoRA, exhibits a distinct update pattern and enhanced learning capacity under data heterogeneity. As we can see, FRLoRA learns more global knowledge throughout the entire communication process compared to FedAvg (non-IID). Even compared to FedAvg (IID), FRLoRA also learns more global knowledge in the early stages (before 50 rounds). This indicates that the update mechanism of FRLoRA facilitates the rapid capture of global knowledge. Moreover, we further found that FRLoRA can significantly mitigate client drift due to its reinitialization mechanism. As a result of these positive effects, FRLoRA improves accuracy from **70.75**% to **75.81**%.

## B   EXPERIMENTAL DETAILS

Our experimental setup follows previous empirical studies (Ye et al., 2024a;b). Detailed information about the datasets used for NLU tasks is provided in Table 6, while Table 7 outlines the datasets for NLG tasks. The training configuration is summarized in Table 8.

### B.1   DATASETS OF NLU TASKS

As shown in Table 6, the NLU experiments are conducted on four datasets, involving different tasks:

- **RTE** (Wang et al., 2019): A two-class classification task based on news and Wikipedia text, where the goal is to determine whether one sentence entails another.

- **COLA** (Wang et al., 2019): It consists of English sentences annotated with acceptability judgments, indicating whether each sentence is grammatically correct based on linguistic theory.

Table 8: **Detailed information of training configuration.**

| Configuration | Batch size | $r$ | $\alpha$ | Rounds | Local steps | Learning rate |
|---|---|---|---|---|---|---|
| RTE | 32 | 16 | 32 | 200 | 10 | 2e-4 |
| COLA | 32 | 16 | 32 | 200 | 30 | 2e-4 |
| QNLI | 32 | 16 | 32 | 200 | 30 | 2e-4 |
| 20NG | 32 | 16 | 32 | 200 | 30 | 2e-4 |
| MetaMathQA | 16 | 32 | 64 | 200 | 10 | 5e-4 |
| Alpaca-GPT4 | 16 | 32 | 64 | 200 | 10 | 5e-4 |
| Fed-Aya | 4 | 16 | 32 | 200 | 10 | 2e-4 |
| Fed-ChatbotIT | 4 | 16 | 32 | 100 | 5 | 2e-4 |
| Fed-WildChat | 4 | 16 | 32 | 100 | 10 | 2e-4 |

- **QNLI** (Wang et al., 2019): A sentence pair classification task where the goal is to determine if a context sentence from a paragraph contains the answer to a given question.
- **20NG** (Lang, 1995): A text classification task to categorize news documents into 20 different newsgroups.

The 20NG dataset contains 20 categories, while the other three datasets consist of two categories. For all datasets, we partition their training set into 5 clients using Dirichlet distribution sampling $(D_k \sim Dir(\beta))$, where a smaller $\beta$ indicates higher data heterogeneity. By default, $\beta$ is set to 0.5. All clients participate in the training process during each round. For evaluation, we report the accuracy (%) on the validation set of RTE, COLA, and QNLI, and Matthews correlation (%) on the test set of 20NG. # Train and # Test represent the number of training and testing samples.

## B.2 DATASETS OF NLG TASKS

**Training Datasets.** Table 7 provides detailed information on five training datasets for various NLG tasks, including:

- **MetaMathQA** (Yu et al., 2023): A dataset containing 395k question answering pairs for math problem solving, augmented from the training sets of GSM8K (Cobbe et al., 2021) and MATH (Yu et al., 2024).
- **Alpaca-GPT4** (Peng et al., 2023): A collection of 52k English instruction-following examples about general knowledge generated by GPT-4, used to fine-tune large language models (LLMs).
- **Fed-Aya** (Ye et al., 2024a): A multilingual instruction tuning dataset with 38 clients and 25,513 samples, including 6 high-resource and 2 low-resource languages.
- **Fed-ChatbotIT** (Ye et al., 2024a): It is a collection of human-annotated preference data from 237 clients with 6166 data samples that capture diversities of realistic use cases in single-turn query of LLMs.
- **Fed-WildChat** (Ye et al., 2024a): It is a collection of multilingual conversations between humans and ChatGPT, containing 100 clients with 52,703 data samples, representing real-world multi-turn interactions between humans and chatbots.

For MetaMathQA and Alpaca-GPT4, we select 20k samples and partition them in an IID manner, with 10 and 20 clients, respectively. At each round, 2 clients are randomly selected to participate in the training. Fed-Aya, Fed-ChatbotIT, and Fed-WilChat are real-world benchmarks with data heterogeneity constructed by Ye et al. (2024a), where we randomly select 4, 10, and 5 clients per round for training. Additionally, we fine-tune LLMs using instruction tuning with Alpaca's instruction template (Taori et al., 2023).

**Evaluation Benchmarks.** We evaluate fine-tuned LLMs on the following test benchmarks:

- **GSM8K** (Cobbe et al., 2021) & **Math** (Yu et al., 2024): Both of them are question-answering datasets focused on math problem solving. We use their test sets for evaluation.

- **MT-Bench** (Zheng et al., 2024): A benchmark designed to evaluate both one-turn (MT-1) and two-turn (MT-2) conversational capabilities of language models across various tasks, including writing, roleplay, reasoning, math, coding, extraction, STEM, and humanities.

- **Vicuna** (Chiang et al., 2023): This benchmark evaluates the one-turn instruction-following capability of language models, covering a range of topics including coding, writing, math, counterfactual reasoning, Fermi problems, common sense, and roleplay.

- **Ref-GPT4** (Ye et al., 2024a): It serves as an in-domain evaluation benchmark, where 50 unseen data samples are randomly selected as the test set. For instance, when a model is fine-tuned on the Fed-Aya dataset, the unseen samples are drawn from Fed-Aya for evaluation.

For GSM8K and Math, we use accuracy (%) to evaluate the performance of LLMs in solving math problems. For the other three benchmarks, we input the outputs of the LLMs into GPT-4 using instruction templates (Ye et al., 2024a), and GPT-4 score them on a scale of 1 to 10, as shown in Table 12 and 13. The prompt template used in GPT-4 Judger from Ye et al. (2024a;b).

## B.3 BASELINES

To demonstrate the effectiveness of our approach, we employ several state-of-the-art heterogeneous FL methods as baselines:

- **FedAvg** (McMahan et al., 2017): A pioneering algorithm that trains models on local datasets and updates the global model by averaging the local model parameters.

- **FedProx** (Li et al., 2020b): It guides local training by introducing a proximity term between the local model and global model parameters.

- **SCAFFOLD** (Karimireddy et al., 2020): During local training, it corrects local gradients by introducing control variates.

- **FedAvgM** (Hsu et al., 2019): It updates the global model through the momentum update mechanism.

- **FedAdagrad** & **FedYogi** & **FedAdam** (Reddi et al., 2021): They integrate adaptive optimization methods into FL, with the difference lying in the use of different computation strategies to obtain the global update. This global update is then applied to the global model through the momentum update mechanism.

- **FFA-LoRA** (Sun et al., 2024): It addresses privacy concerns and mitigates data heterogeneity issues by fixing the $A$ matrix for each client and only optimizing the $B$ matrix.

Aside from FFA-LoRA, the remaining methods are centered on FFT-based FL, which we integrate with LoRA. We utilize the hyperparameters specified in Ye et al. (2024a) for these methods.

## B.4 MODELS

We use RoBERTa-base and LLaMA-2-7B as the base models for NLU and NLG tasks, respectively. The total number of model parameters (# Param.base) and trainable parameters (# Param.trainable) are shown in the Table 9.

Table 9: **Illustration of the number of model parameters.**

|  | Configure | # Param.base | # Param.trainable |
|---|---|---|---|
| RoBERTa-base | $r = 16 \; \& \; \alpha = 32$ | 125.82M | 1.77M |
| LLaMA-2-7B | $r = 16 \; \& \; \alpha = 32$ | 6738M | 2.09M |

Table 10: **Experimental results under various ranks**, where **MetaMathQA** is used as the training dataset. **Avg.** is the average result of corresponding metrics, and the best results are marked in **bold**.

| Method | r = 16 | | | r = 32 | | | r = 64 | | |
|---|---|---|---|---|---|---|---|---|---|
| | GSM8K | Math | Avg. | GSM8K | Math | Avg. | GSM8K | Math | Avg. |
| FedAvg | 32.67 | 4.64 | 18.65 | 34.95 | 4.48 | 19.71 | 37.45 | 5.38 | 21.41 |
| FedProx | 32.29 | 4.32 | 18.30 | 35.40 | 4.66 | 20.03 | 36.39 | 4.98 | 20.68 |
| SCAFFOLD | 32.97 | 4.70 | 18.84 | 35.78 | 5.08 | 20.43 | 32.37 | 4.64 | 18.50 |
| FedAvgM | 32.44 | 4.42 | 18.43 | 34.79 | 4.64 | 19.71 | 35.57 | 4.72 | 20.14 |
| FedAdagrad | 28.65 | 4.18 | 16.41 | 29.64 | 4.06 | 16.85 | 31.76 | 4.86 | 18.31 |
| FedYogi | 30.32 | 4.00 | 17.16 | 30.09 | 4.04 | 17.06 | 33.96 | 4.70 | 19.33 |
| FedAdam | 31.23 | 4.14 | 17.68 | 31.84 | 4.12 | 17.98 | 34.26 | 5.18 | 39.44 |
| FFA-LoRA | 25.17 | 3.60 | 14.38 | 28.05 | 3.78 | 15.91 | 31.00 | 4.50 | 17.75 |
| **FRLoRA (Ours)** | **39.57** | **5.60** | **22.58** | **44.27** | **5.22** | **24.74** | **45.56** | **6.88** | **26.22** |

Table 11: **Experimental results under various LLMs**, where **MetaMathQA** is used as the training dataset. **Avg.** is the average result of corresponding metrics, and the best results are marked in **bold**.

| Method | Qwen2-1.5B | | | Gemma-2B | | | LLaMA-2-7B | | |
|---|---|---|---|---|---|---|---|---|---|
| | GSM8K | Math | Avg. | GSM8K | Math | Avg. | GSM8K | Math | Avg. |
| FedAvg | 16.30 | 1.66 | 8.98 | 36.08 | 15.00 | 25.54 | 34.95 | 4.48 | 19.71 |
| FedProx | 16.98 | 2.20 | 9.59 | 35.63 | 14.30 | 24.96 | 35.40 | 4.66 | 20.03 |
| SCAFFOLD | 14.02 | 1.56 | 7.79 | 37.07 | 14.52 | 25.79 | 35.78 | 5.08 | 20.43 |
| FedAvgM | 10.31 | 0.76 | 5.53 | 36.69 | 13.80 | 25.24 | 34.79 | 4.64 | 19.71 |
| FedAdagrad | 43.82 | 2.92 | 23.37 | 34.64 | 14.14 | 24.39 | 29.64 | 4.06 | 16.85 |
| FedYogi | 0.83 | 0.40 | 0.61 | 34.49 | 13.84 | 24.16 | 30.09 | 4.04 | 17.06 |
| FedAdam | 1.28 | 0.48 | 0.88 | 34.57 | 13.46 | 24.01 | 31.84 | 4.12 | 17.98 |
| FFA-LoRA | 36.01 | 4.24 | 20.12 | 32.90 | 14.28 | 23.59 | 28.05 | 3.78 | 15.91 |
| **FRLoRA (Ours)** | **52.08** | **17.34** | **34.71** | **42.30** | **15.24** | **28.77** | **44.27** | **5.22** | **24.74** |

## C    ADDITIONAL EXPERIMENTS

### C.1    VARIOUS RANKS

In this section, we investigate the impact of varying ranks. Specifically, we adjust the rank to $\{16, 32, 64\}$ while keeping the scale factor $\frac{\alpha}{r}$ fixed at 2, with all other experimental settings remaining constant. The experimental results on MetaMathQA are shown in Table 10. As observed, most methods exhibit a consistent trend: higher ranks yield improved performance. This is attributed to the greater learning capacity of larger ranks, enabling more effective representation of knowledge across diverse clients. These findings corroborate our claim in § 1. Moreover, FRLoRA consistently outperforms all other methods at different ranks, indicating its scalability and robustness to rank variations. Besides, we also observe that FRLoRA ($r$=16) achieved even higher performance than FedAvg ($r$=64), demonstrating its effectiveness in addressing the constrained parameter space.

### C.2    VARIOUS LLMS

We further evaluate the effectiveness of our method across different LLMs. To this end, we selected two models with varying architectures and sizes: Qwen2-1.5B (Yang et al., 2024) and Gemma-2B (Team et al., 2024), and conducted additional experiments on MetaMathQA. The $r$ and $\alpha$ were set to 32 and 64, respectively, while keeping all other experimental settings unchanged. Specifically,

FedYogi and FedAdam achieved poor results, with average accuracies (Avg.) of only **0.61**% and **0.88**%, respectively. On the other hand, FedAdagrad exhibited significant performance improvements compared to other baselines, which contrasts with its behaviour on Gemma-2B and Llama-2-7B. Notably, FRLoRA consistently outperforms all baselines across the three LLMs, even achieving superior performance with Qwen2-1.5B, a smaller model compared to the other two. This demonstrates the strong generalizability of our method across different LLM architectures and scales.

Table 12: **Examples of judgement by GPT-4 on Vicuna.**

**Question:** *How many text messages are sent globally in a minute? Try to explain your answer. Your explanation should take the reader through your reasoning step-by-step.*

**FedAvg:** *Let's take a look at the data and see what we can uncover. According to a report by The Guardian, there were 47.9 billion text messages sent globally in 2019.*

*Now, let's break it down:*

*- The report states there were 4.6 trillion text messages sent globally in 2019. This equals to 12.6 billion text messages sent per day.*

*- According to a report by GlobalData, there were 6.18 trillion texts sent in 2019, which is roughly 17 billion texts per day.*

*To explain this further, we can look at the data from the reports and consider that the reports are from 2019, and that it is likely that the number of text messages sent has increased since then.*

*In the end, there is no definitive answer as to how many text messages are sent globally in a minute, but we can estimate that on average, around 12-17 billion text messages are sent in a minute. The figure could be higher or lower depending on the source and the time period considered.*

**GPT-4 Judger:** *The assistant's response is detailed and well-reasoned, but there is* a significant error *in the calculations. The assistant correctly cites data from The Guardian and GlobalData, but the conversion from daily to minute text messages is incorrect. If there are 12.6 billion to 17 billion texts sent per day, this does not translate to the same number being sent per minute. The assistant should have divided the daily figure by the number of minutes in a day (1440) to get the correct answer. This error* significantly impacts the accuracy and relevance *of the response.* **Therefore, the rating is lower. Rating: [[3]]**

**FRLoRA:** *Unfortunately, there is no definitive answer to the number of text messages sent globally in a minute, as this figure would require the collection of data from all telecommunication companies and mobile phone operators across the globe. It is difficult to estimate the number of texts sent per second, let alone per minute. However, according to Statista, in 2022, the number of text messages sent in the U.S. alone was 114.7 billion per month, which equals to about 2,978 million per day. This means that a large number of text messages are being sent globally every minute, but it is difficult to determine an exact figure.*

*To explain further, the number of text messages sent in a minute would depend on several factors such as the population of the countries, the frequency of texting, and the number of mobile phone subscribers. For example, in countries with a large population and a high rate of mobile phone penetration, such as India and China, the number of text messages sent per minute would be significantly higher than in countries with smaller populations or lower mobile phone usage, like Sweden or Norway. Additionally, the number of text messages sent would also vary depending on the time of the day, as people tend to send more texts during peak hours, such as during office hours or after work.*

*In conclusion, the number of text messages sent globally in a minute is not a fixed figure but varies depending on several factors. While we cannot give an exact number, we can say that texting is a popular form of communication and a large number of texts are being sent globally every minute.*

**GPT-4 Judger:** *The assistant's response is* highly informative and detailed. *It explains why it's difficult to provide an exact number of text messages sent globally per minute, citing the need for data from all telecommunication companies and mobile phone operators worldwide. The assistant also provides a statistic from Statista to give a sense of the scale of text messaging in the U.S., which helps to illustrate the global scale. The assistant further explains the factors that would influence the number of texts sent per minute, such as population size, mobile phone usage, and time of day. The assistant's response is* accurate, relevant, *and provides* a good depth *of information, even though it couldn't provide a specific number.* **Rating: [[9]]**

Table 13: **Examples of judgement by GPT-4 on Vicuna.**

**Question:** *What are the main differences between Python and JavaScript programming languages?*

**FedAvg:** *Python and JavaScript are two popular programming languages that have many similarities, but there are some key differences between them. Here are some of the main ones:*

*1. Python is an object-oriented programming language, while JavaScript is a functional programming language. Object-oriented programming is used to organize and structure code by grouping related data and methods into objects. Functional programming, on the other hand, focuses on functions and their interactions with other functions rather than on objects and classes. 2. Python is a high-level programming language, while JavaScript is a low-level programming language. This means that Python requires less manual memory management and handles memory allocation automatically, while JavaScript requires the programmer to manage memory manually. 3. Python has a built-in collection of standard libraries, while JavaScript does not. Python's standard library is extensive, making it easier for programmers to write code for common tasks without having to reinvent the wheel. JavaScript, on the other hand, requires the programmer to write their own code or use third-party libraries to perform the same tasks. 4. Python has a syntax that is more readable and natural than JavaScript, making it easier for programmers to understand and write code in Python. JavaScript has a syntax that is more compact and concise, which can make it easier to write code quickly. 5. Python is an interpreted programming language, while JavaScript is a compiled programming language. This means that JavaScript code is first translated into machine code before it is executed, while Python code is executed directly by the interpreter. 6. Python is a dynamic programming language, while JavaScript is a static programming language. This means that Python code can be executed at any time and the results may change, while JavaScript code must have a specific order and cannot change.*

*Overall, the choice between Python and JavaScript depends on the specific needs and preferences of the programmer. While both languages have their strengths and weaknesses, they are both widely used and have a large developer community.*

**GPT-4 Judger:** *The assistant's response is detailed and provides a comprehensive comparison between Python and JavaScript. However, there are* some inaccuracies in the information. *For instance, JavaScript is not a low-level programming language, it's also high-level like Python. Also, JavaScript is not a compiled language, it's interpreted just like Python. Furthermore, JavaScript is not a purely functional programming language, it supports both object-oriented and functional programming paradigms. Lastly, JavaScript is not a static language, it's dynamic like Python. These inaccuracies* significantly affect the quality of the response. **Therefore, the rating is: [[3]]**

**FRLoRA:** *Python and JavaScript are both popular programming languages used for web development and general-purpose scripting. However, there are several key differences between the two languages:*

*1. Language Typing: Python is a strongly typed language, which means that variables must have a specific data type. JavaScript, on the other hand, is a loosely typed language, meaning that variables can have any data type. 2. Syntax: Python has a more rigid syntax compared to JavaScript. For example, in Python, indentation is used to indicate structure, while in JavaScript, curly braces are used to denote blocks of code. 3. Data Structures: In Python, there are several built-in data structures such as lists, dictionaries, sets, and tuples. JavaScript, on the other hand, has a more limited set of built-in data structures, including arrays and objects. 4. Functions: Functions in Python are first-class citizens, which means that they can be assigned, passed as arguments, and returned from other functions. In JavaScript, functions are second-class citizens, meaning that they can only be passed as arguments and returned from other functions. 5. Object-Oriented Programming: Both languages support Object-Oriented Programming (OOP), but Python has a more explicit focus on it, with features such as classes and inheritance. JavaScript has a more limited support for OOP, with only classes and prototypes. 6. Web Development: Both languages can be used for web development, but Python is typically used for back-end development, while JavaScript is used for front-end development. Python can be used to build web applications using frameworks like Django or Flask, while JavaScript is used to build interactive client-side interfaces using frameworks like React or Angular.*

*Overall, both Python and JavaScript are powerful and versatile languages, but they have different strengths and weaknesses, and they are used for different types of development projects.*

**GPT-4 Judger:** *The assistant's response is* highly informative, accurate, and detailed. *It covers the main differences between Python and JavaScript, including language typing, syntax, data structures, functions, object-oriented programming, and their uses in web development. The assistant also provides examples to clarify the points. The response is* well-structured and easy to understand, *making it* very helpful *for someone trying to understand the differences between these two programming languages.* **Therefore, I would rate this response highly. Rating: [[9]]**

Table 14: **Communication cost per round** on **Alpaca-GPT4** with LLaMA-2-7B, where $r$ is 32. **Cost.P** denotes the communication overhead from participating training clients, **Cost.NP** represents the overhead from non-participating clients, and **Cost.T** indicates the total communication cost.

| Method | Cost.P (MB) | Cost.NP (MB) | Cost.T (MB) | Vicuna | MT-1 | MT-2 | MT-Avg | Avg. |
|---|---|---|---|---|---|---|---|---|
| *Partial Participation Setting* | | | | | | | | |
| FFT-based FL | 13476× 2 | 0 | 26952 | | | *Out of Memory* | | |
| FedAvg | 8.388 × 2 | 0 | 16.77 | 7.925 | 4.650 | 2.025 | 3.346 | 4.486 |
| **FRLoRA** | 8.388 × 2 | 4.194 × 18 | 92.26 | **8.044** | **4.775** | **2.481** | **3.635** | **4.733** |
| *Full Participation Setting* | | | | | | | | |
| FFT-based FL | 13476× 20 | 0 | 269520 | | | *Out of Memory* | | |
| FedAvg | 8.388 × 20 | 0 | 167.76 | 8.039 | 4.833 | 2.458 | 3.645 | 4.743 |
| **FRLoRA** | 8.388 × 20 | 0 | 167.76 | **8.125** | **4.984** | **2.806** | **3.895** | **4.952** |

## C.3 COMMUNICATION OVERHEAD

In the full participation setting, where all clients are involved in training every round, `FRLoRA` incurs no additional overhead, as local models remain consistent through parameter synchronization. However, in the partial participation setting where only a subset of clients participate in training every round, `FRLoRA` introduces extra communication overhead because inactive clients must receive the global low-rank matrix from the server for the parameter synchronization. Since synchronization occurs through the global low-rank matrices, the increased communication cost remains acceptable. As shown in Table 14, the additional overhead introduced by `FRLoRA` per round is 75.49 MB, which accounts for only **0.28**% of the communication cost in FFT-based FL. This demonstrates that LoRA-based FL can significantly reduce the communication overhead when fine-tuning LLMs. Furthermore, `FRLoRA` does not increase the communication cost per client. As the communication channels between each client and the server are independent and parallel, the overhead from inactive clients does not affect overall communication efficiency.

Besides, we further present the performance in the full participation setting. It can be observed that under this setting, `FRLoRA` does not incur any additional communication overhead while maintaining superior performance. This also indicates that the performance improvement in the partial participation setting does not stem from the increased communication overhead caused by inactive clients.

## C.4 EFFICIENCY OF SVD COMPUTATION

We conducted an analysis of the time and memory requirements for the SVD computation in our method. As shown in Table 15, the computational cost of SVD is minimal in terms of memory, with peak memory consumption remaining under 1 MB for both models. While the time cost scales with model size, SVD is performed only once, rendering its overall impact negligible compared to the whole training phase.

Table 15: **Time and peak memory** for the SVD computation.

| Model | Time | Peak Memory |
|---|---|---|
| RoBERTa-base | 1.48 s | 0.25 MB |
| LLaMA-2-7B | 197.82 s | 0.65 MB |

## C.5 ANALYSIS OF CLIENT DRIFT

In this section, we further analyze the effectiveness of our method in mitigating client drift. To achieve this, we calculate the cosine similarity among local updates $\Delta W_k^t$ across different clients. We compare the results of FedAvg and `FRLoRA` on RTE, using the same number of training rounds to ensure a fair evaluation. The results show that the cosine similarity between clients in FedAvg

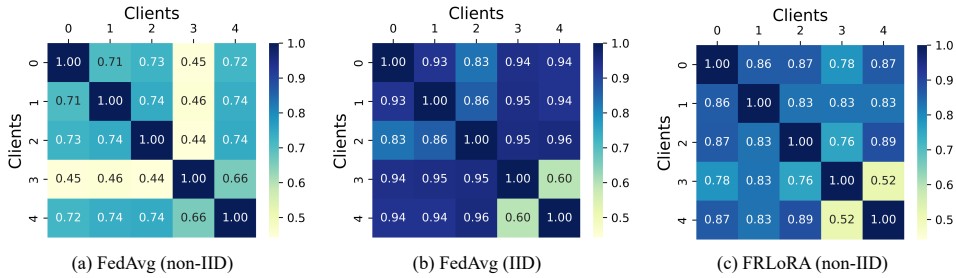

(a) FedAvg (non-IID)    (b) FedAvg (IID)    (c) FRLoRA (non-IID)

Figure 3: **Illustration of the cosine similarity distribution** among local updates across different clients on RTE.

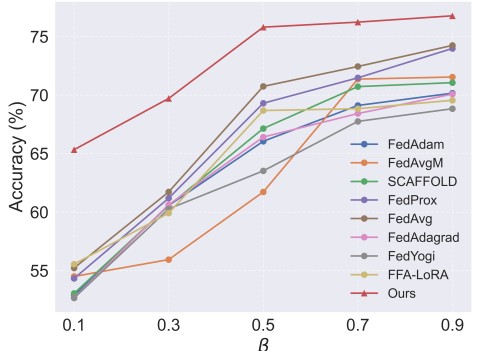

Figure 4: **Accuracy comparison** of different FL methods in terms of various $\beta$ on RTE.

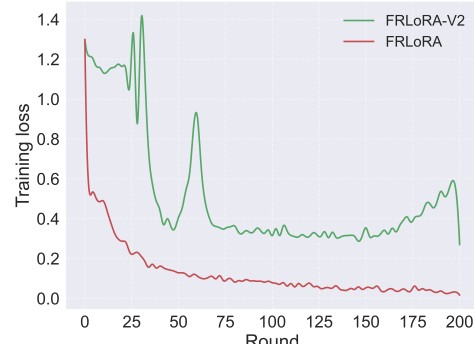

Figure 5: **Illustration of global training loss** versus communication rounds on RTE.

(IID) (see Figure 3 (b)) is distributed significantly higher than in FedAvg (non-IID) (see Figure 3 (a)), and FRLoRA (non-IID) (see Figure 3 (c)) exhibits a higher distribution compared to FedAvg (non-IID). This further demonstrates the effectiveness of FRLoRA in reducing client drift, promoting more consistent model convergence under data heterogeneity.

### C.6  EFFECT OF DATA HETEROGENEITY

In this section, we investigate the impact of data heterogeneity on our method by varying the $\beta$ of the Dirichlet distribution across the values $0.1, 0.3, 0.5, 0.7, 0.9$ on the RTE dataset, where smaller values of $\beta$ correspond to higher data heterogeneity. As shown in Figure 4, we can observe that the accuracy of all methods increases with the increase of $\beta$, and FRLoRA significantly outperforms the other methods at different values of $\beta$. Moreover, the gap is larger when $\beta$ is small, indicating the effectiveness of FRLoRA in addressing data heterogeneity.

### C.7  ANALYSIS OF INITIALIZATION

To further investigate the impact of initialization on FRLoRA, we compared the convergence behavior of FRLoRA-v2 with our method. As stated in §4.4, FRLoRA-v2 is a variant that reinitializes the low-rank matrices to zero and Gaussian noise in each round. The results in Figure 5 demonstrates that FRLoRA-v2 exhibits slower and less stable convergence compared to FRLoRA. This demonstrates the effectiveness of FRLoRA's initialization strategy in addressing the convergence challenges associated with standard initialization methods, enabling faster and more stable convergence. This facilitates $\Delta W^t$ in effectively capturing global knowledge.

## D  ADDITIONAL THEORETICAL ANALYSIS

In this section, we present additional theoretical analysis on the rank of the global update.

To begin with, we define the global update of FedAvg with LoRA as,

$$\Delta\widetilde{\boldsymbol{W}}^T_{FedAvg} = \boldsymbol{B}^T_G \boldsymbol{A}^T_G.$$

Here, $\boldsymbol{B}^T_G$ is a $d_1 \times r$ matrix and $\boldsymbol{A}^T_G$ is a $r \times d_2$ matrix. Based on Eq.(12), we have,

$$rank(\Delta\widetilde{\boldsymbol{W}}^T_{FedAvg}) \leq \min(rank(\boldsymbol{B}^T_G), rank(\boldsymbol{A}^T_G)) \leq r.$$

Then, we define the global update of FRLoRA as,

$$\Delta\widetilde{\boldsymbol{W}}^T_{FRLoRA} = \Delta\boldsymbol{W}^1 + \Delta\boldsymbol{W}^2 + \ldots + \Delta\boldsymbol{W}^T$$

$$= (\boldsymbol{B}^1_G\boldsymbol{A}^1_G - \boldsymbol{B}^0_G\boldsymbol{A}^0_G) + (\boldsymbol{B}^2_G\boldsymbol{A}^2_G - \boldsymbol{B}^0_G\boldsymbol{A}^0_G) + \ldots + (\boldsymbol{B}^T_G\boldsymbol{A}^T_G - \boldsymbol{B}^0_G\boldsymbol{A}^0_G)$$

$$= -T\boldsymbol{B}^0_G\boldsymbol{A}^0_G + \boldsymbol{B}^1_G\boldsymbol{A}^1_G + \boldsymbol{B}^2_G\boldsymbol{A}^2_G + \ldots + \boldsymbol{B}^T_G\boldsymbol{A}^T_G.$$

Obviously, $\Delta\widetilde{\boldsymbol{W}}^T_{FRLoRA}$ can be rewritten as,

$$\Delta\widetilde{\boldsymbol{W}}^T_{FRLoRA} = [-T\boldsymbol{B}^0_G; \boldsymbol{B}^1_G; \boldsymbol{B}^2_G; \ldots; \boldsymbol{B}^T_G] \times [\boldsymbol{A}^0_G; \boldsymbol{A}^1_G; \boldsymbol{A}^2_G; \ldots; \boldsymbol{A}^T_G].$$

Here, the size of $[-T\boldsymbol{B}^0_G; \boldsymbol{B}^1_G; \boldsymbol{B}^2_G; \ldots; \boldsymbol{B}^T_G]$ is $d_1 \times r(T + 1)$, and the size of $[\boldsymbol{A}^0_G; \boldsymbol{A}^1_G; \boldsymbol{A}^2_G; \ldots; \boldsymbol{A}^T_G]$ is $r(T + 1) \times d_2$. Thus, we have,

$$rank(\Delta\widetilde{\boldsymbol{W}}^T_{FRLoRA}) \leq \min(r(T + 1), d_1, d_2),$$

and

$$rank(\Delta\widetilde{\boldsymbol{W}}^T_{FRLoRA}) \geq rank(\Delta\widetilde{\boldsymbol{W}}^T_{FedAvg}).$$

Finally, we note that $rank(\Delta\widetilde{\boldsymbol{W}}^T_{FRLoRA})$ can be greater than $rank(\Delta\widetilde{\boldsymbol{W}}^T_{FedAvg})$ only except that $\boldsymbol{B}^0_G\boldsymbol{A}^0_G, \boldsymbol{B}^1_G\boldsymbol{A}^1_G, \ldots, \boldsymbol{B}^T_G\boldsymbol{A}^T_G$ are completely linearly independent.

