# OpenReview forum: "Federated Residual Low-Rank Adaptation of Large Language Models"
_ICLR.cc/2025/Conference — ICLR 2025 Poster_

### Official Review · Reviewer_Ddf1 · 2024-10-30

**Soundness:** 2
**Presentation:** 3
**Contribution:** 2
**Rating:** 8
**Confidence:** 4

**Summary:**

In this paper, the author proposes a novel framework named federated residual low-rank adaption to effectively fine-tune the pretrained large language model in a privacy-preserving manner. The proposed method addresses both the intrinsic problem caused by constrained parameter space and the extrinsic problem caused by the drift among clients. Extensive experiments on several datasets have shown the effectiveness of the proposed method.

**Strengths:**

1. The idea of enlarging the parameter space when finetuning large laugunge model in federated learning is promising.
2. The paper is well organized and the related works are thoroughly summarized.
3. The discussion of the weakness of exsiting related works and the novelty of the proposed method are clear.
4. The author conducts comprehensive experiments to demonstrate the effectiveness of the proposed method, which is convincing.

**Weaknesses:**

1. In Eqs. (6) and (7), the author initializes the local low-rank matrices by the low-rank matrices reconstructed from the top singular values, while remaining the residual part as the global model . In my opinion, this makes the low-rank matrices retain the most information in the pretrained model, while the information remained in the global model can be significanly reduces.  However, in FL, the global model should contain more generalized information from the pre-trained model, which is contrast to the above parameter decoupling operation. So I am curious that whether this operation can reduce the generalization capacity or convergence speed of FL?

2. The motivation of the proposed low-rank matrice initilization lacks further discussion. Can the author provide more detailed discussion about this by such as landscape visualization?

3. Why the global model are initialized by the residual matrice? It seems that directly using the pre-trained model parameters as the global model can provide a more generalized initialization during the FL training. Can the author provide a more detailed discussion about this?

4. In Eqs. (13) and (14),  the author claims that by using residual accumulation, the model fine-tuned space can be expanded. However, Eq. (14) seems to only give a upper bound of the summed matrices rather than increasing its lowe bounder of the summed matrices. So it can not guarantee that the rank of the summed matrices can be increased by such a residual accumulation operation. Can the author provide a more in-depth discussion about this?

5. Actually, the standard PEFT by LoRa in FL can also be formulated as Eq. (13), with $\delta W^t$ as the global update of the LoRa parts at each global round. So what is the real differences between the standard method and the proposed FRLoRA?

**Questions:**

The items listed in the weaknesses. If the author can address or partially address these problems, I will be pleasure to improve my score.

---

> ### Author Response · Authors · 2024-11-23
> **Official Comment by Authors (1/3)**
>
> We thank the reviewer for his/her constructive comments and provide our point-wise replies as follows.
>
> > **Q1:** Does Eqs.(6) and (7) result in Information loss?
>
> We completely **AGREE** that $\boldsymbol{A}_G^0$ and $\boldsymbol{B}_G^0$ retain the **MOST** information in the pretrained model. Actually, they are merged back into the global model (see Eq.(11)), and this does **NOT** result in information loss:
>
> (1) **Reinitialization at Each Round:** At the start of each communication round, the low-rank matrices are reinitialized to $\boldsymbol{A}_G^0$ and $\boldsymbol{B}_G^0$, ensuring the critical information is always ***preserved***.
>
> (2) **Final Integration:** After training, $\boldsymbol{A}_G^0$ and $\boldsymbol{B}_G^0$ are ***merged*** back into the global model (Eq.(11)), fully ***restoring*** all retained information.
>
>
> > **Q2:** Motivation of FRLoRA's initialization.
>
> We clarify it as follows:
>
> (1) The standard method, where $\boldsymbol{B}$ is set to 0 and $\boldsymbol{A}$ is initialized with Gaussian noise, often struggles with convergence. If FRLoRA is reinitialized to such matrices in each round, it will severely hinder the convergence of the local models, thereby degrading the knowledge in the residual update $\Delta \boldsymbol{W}^t$. We have added a visualization of the loss landscape in the revision (Appendix C.7 Figure 5). The results demonstrates the effectiveness of FRLoRA's initialization strategy in addressing the convergence challenges associated with standard initialization methods, enabling **faster** and more **stable** convergence.
>
> (2) Moreover, under data heterogeneity, standard initialization causes inconsistent convergence due to the varying data distributions across clients, further exacerbating client drift. FRLoRA's reinitialization ensures that each round of local optimization begins with a consistent principal singular value space, helping to mitigate client drift. Our ablation results (Section 4.4) on different datasets and further analysis in the revision (Appendix C.5 Figure 3), which confirm its effectiveness.
>
> > **Q3:** Why the global model are initialized by the residual matrice?
>
>
> If the global model in Eq.(7) is initialized directly using the pretrained model, i.e., $\hat{\boldsymbol{W}}^0 =\boldsymbol{W}^0$, the model parameters $\widetilde{\boldsymbol{W}}^0$ in Eq.(11) at the initial stage becomes:
>
> $$
> \begin{aligned}
>     \widetilde{\boldsymbol{W}}^0 &= \hat{\boldsymbol{W}}^0 + \boldsymbol{B}^{0}_G \boldsymbol{A}^{0}_G \\\\
>     &= \boldsymbol{W}^0 + \boldsymbol{U}[:r]\sqrt{\boldsymbol{S}[:r]} \times  \sqrt{\boldsymbol{S}[:r]} \boldsymbol{V}[:r] \\\\
>     &\neq \boldsymbol{W}^0
> \end{aligned}
> $$
> This indicates that, at the initial stage, merging the initialized low-rank matrices back into the global model does not revert it to the pretrained model. Therefore, we should initialize the global model with residual matrices instead of the pretrained model, which ensures that $\widetilde{\boldsymbol{W}}^0$ and $\boldsymbol{W}^0$ remain consistent in the initial stage.

---

> ### Author Response · Authors · 2024-11-23
> **Official Comment by Authors (2/3)**
>
> > **Q4:** Explanation of residual accumulation.
>
> We discuss this both theoretically and empirically.
>
> (1) **Theoretically**, the upper bound on the rank of the global update parameter space in FRLoRA is $\min(rT, d_1, d_2)$, which is significantly **larger** than the upper bound $r$ in FedAvg. The iterative residual accumulation mechanism provides this **higher** upper bound, though we also acknowledge that it cannot strictly increase the rank as $T$ increases. However, a higher upper bound typically means a **larger** potential maximum value under the same conditions. This indicates that FRLoRA has the **ability** to explore a larger updating parameter space to capture more complex structures, allowing for **better** representation of the diverse knowledge learned from different clients.
>
> (2) **Empirically,** we have provided results with different values of $r$ in our submitted paper (Table 10). The results are presented below:
>
> | Method       | GSM8K (r=16) | Math (r=16) | Avg. (r=16) | GSM8K (r=32) | Math (r=32) | Avg. (r=32) | GSM8K (r=64) | Math (r=64) | Avg. (r=64) |
> |--------------|--------------|-------------|-------------|--------------|-------------|-------------|--------------|-------------|-------------|
> | FedAvg       | 32.67        | 4.64        | 18.65       | 34.95        | 4.48        | 19.71       | 37.45        | 5.38        | 21.41       |
> | FedProx      | 32.29        | 4.32        | 18.30       | 35.40        | 4.66        | 20.03       | 36.39        | 4.98        | 20.68       |
> | SCAFFOLD     | 32.97        | 4.70        | 18.84       | 35.78        | 5.08        | 20.43       | 32.37        | 4.64        | 18.50       |
> | FedAvgM      | 32.44        | 4.42        | 18.43       | 34.79        | 4.64        | 19.71       | 35.57        | 4.72        | 20.14       |
> | FedAdagrad   | 28.65        | 4.18        | 16.41       | 29.64        | 4.06        | 16.85       | 31.76        | 4.46        | 18.31       |
> | FedYogi      | 30.32        | 4.00        | 17.16       | 30.09        | 4.04        | 17.06       | 33.96        | 4.40        | 19.33       |
> | FedAdam      | 31.23        | 4.14        | 17.68       | 31.84        | 4.12        | 17.98       | 34.26        | 5.16        | 39.44       |
> | FFA-LoRA     | 25.17        | 3.60        | 14.38       | 28.05        | 3.78        | 15.91       | 31.00        | 4.50        | 17.75       |
> | **FRLoRA (Ours)** | **39.57**    | **5.60**    | **22.58**    | **44.27**    | **5.22**    | **24.74**    | **45.56**    | **6.88**    | **26.22**    |
>
> As shown, FRLoRA ($r$ = 16) outperforms FedAvg ($r$=16) and even achieves ***higher*** performance than FedAvg (r=64). Besides, we observed that when training FRLoRA ($r=16$) on MetaMathQA after 100 rounds, the ***average*** rank of residual accumulation at each layer reached **82**. These results ***strongly*** demonstrate the effectiveness of our method in expanding the parameter space of global updates.

---

> ### Author Response · Authors · 2024-11-23
> **Official Comment by Authors (3/3)**
>
> > **Q5:** Differences between the standard method and FRLoRA in Eq. (13).
>
>
> The standard LoRA in FL can **NOT** be formulated as Eq.(13). If we want to express the global update of the LoRa parts in the form of Eq. (13), it **only** can be written as:
>
>
> $$
> \begin{aligned}
> \boldsymbol{A}_G^T &= \boldsymbol{A}_G^0 + \boldsymbol{A}_G^1 - \boldsymbol{A}_G^0 + \boldsymbol{A}_G^2 - \boldsymbol{A}_G^1 + \ldots + \boldsymbol{A}_G^T - \boldsymbol{A}_G^{T-1} \\\\
>  &= \boldsymbol{A}_G^0 + \Delta \boldsymbol{A}^1 +  \Delta \boldsymbol{A}^2 + \ldots +  \Delta \boldsymbol{A}^T \\\\
>  \end{aligned}
> $$
>
> $$
> \begin{aligned}
> \boldsymbol{B}_G^T &= \boldsymbol{B}_G^0 + \boldsymbol{B}_G^1 - \boldsymbol{B}_G^0 + \boldsymbol{B}_G^2 - \boldsymbol{B}_G^1 + \ldots + \boldsymbol{B}_G^T - \boldsymbol{B}_G^{T-1} \\\\
>  &= \boldsymbol{B}_G^0 + \Delta \boldsymbol{B}^1 +  \Delta \boldsymbol{B}^2 + \ldots +  \Delta \boldsymbol{B}^T
>  \end{aligned}
> $$
>
> We can observe that $\Delta\boldsymbol{A}^{t}$ and $\Delta\boldsymbol{A}^{t-1}$, as well as $\Delta\boldsymbol{B}^{t}$ and $\Delta\boldsymbol{B}^{t-1}$, are **NOT** independent. They share **common terms**, as the standard LoRA in FL only updates the low-rank matrices and uses them as the initialization for the next training round. Since these common terms can be **merged**, there is **NO** iterative accumulative effect as shown in Eq.(13). Therefore, the final fine-tuned global model can **only** be expressed as Eq. (4):
>
> $$
> \widetilde{\boldsymbol{W}}^T = \boldsymbol{W}^0 + \Delta\boldsymbol{W}^T = \boldsymbol{W}^0 + \boldsymbol{B}_G^T\boldsymbol{A}_G^T
> $$
>
> In contrast, the global update process of FRLoRA can be expressed as:
>
> $$
> \begin{aligned}
>     \widetilde{\boldsymbol{W}}^{T} &= \hat{\boldsymbol{W}}^T + \boldsymbol{B}^{0}_G \boldsymbol{A}^{0}_G \\\\
>     &= \hat{\boldsymbol{W}}^0 + \Delta\boldsymbol{W}^1 + \Delta\boldsymbol{W}^2 + \ldots + \Delta\boldsymbol{W}^T + \boldsymbol{B}^{0}_G \boldsymbol{A}^{0}_G \\\\
>     &= \boldsymbol{W}_0 - \boldsymbol{B}_G^0 \boldsymbol{A}_G^0 + \Delta\boldsymbol{W}^1 + \Delta\boldsymbol{W}^2 + \ldots + \Delta\boldsymbol{W}^T + \boldsymbol{B}^{0}_G \boldsymbol{A}^{0}_G \\\\
>     &=\boldsymbol{W}_0 + \Delta\boldsymbol{W}^1 + \Delta\boldsymbol{W}^2 + \ldots + \Delta\boldsymbol{W}^T \\\\
>     &= \boldsymbol{W}^0 + \boldsymbol{B}_G^1\boldsymbol{A}_G^1 - \boldsymbol{B}_G^0\boldsymbol{A}_G^0 +  \boldsymbol{B}_G^2\boldsymbol{A}_G^2 - \boldsymbol{B}_G^0\boldsymbol{A}_G^0 + \ldots+\boldsymbol{B}_G^T\boldsymbol{A}_G^T - \boldsymbol{B}_G^0\boldsymbol{A}_G^0
> \end{aligned}
> $$
>
> Given that FRLoRA reinitializes the low-rank matrices in each training round, $\Delta\boldsymbol{W}^{t}$ and $\Delta\boldsymbol{W}^{t-1}$ are **independent** and can **NOT** be merged. Additionally, FRLoRA **directly** applies these global updates to the global model.  As a result, the global update of FRLoRA has an iterative accumulative effect, which allows gobal updates to occur in a higher-rank parameter space, thereby effectively capturing global knowledge.
>
>
> Besides, $\Delta \boldsymbol{A}^1, \ldots, \Delta\boldsymbol{A}^T$ are all $r\times d_2$ matrices, and when summing $T$ such $r \times d_2$ matrices, the resulting matrix's rank will not exceed $r$. Although $\Delta W^t$ is a $d_1 \times d_2$ matrix with rank $r$, when summing $T$ such matrices, the upper bound of rank should be $\min(rT, d_1, d_2)$, which can greater than $r$.

---

> > ### Author Response · Authors · 2024-11-25
> > **Look forward to your post-rebuttal feedback**
> >
> > As the discussion period draws to a close soon, we extend our sincere gratitude to you for the valuable time and insightful comments.
> >
> > In our previous response, we have carefully studied your comments and made detailed responses summarized below:
> >
> > 1. Clarified the issue of information loss in Eqs.(6) and (7).
> > 2. Provided further explanation on the motivation of FRLoRA's initialization.
> > 3. Explained why the global model should be initialized the residual matrice.
> > 4. Provided further explanation on the residual accumulation in Eq.(13).
> > 5. Explained the differences between the standard method and FRLoRA in Eq.(13).
> >
> > We sincerely hope our responses have effectively addressed your concerns. If you have any remaining questions or require further clarification, please do not hesitate to let us know, and we would be glad to provide further explanations
> >
> > Thank you again for your efforts in reviewing our work.

---

> > ### Comment · Reviewer_Ddf1 · 2024-11-25
> >
> > The authors have partially addressed my concern and I improved my score as a result. However, I still have some concerns about the theretical results of residual accumulation.

---

> > > ### Author Response · Authors · 2024-11-25
> > > **Official Comment by Authors**
> > >
> > > Thank you for your prompt response. We will immediately address your remaining concerns, such as those regarding the lower bounds. We are highly committed to resolving all your concerns to your satisfaction.

---

> ### Author Response · Authors · 2024-11-26
> **Official Comment by Authors**
>
> Thanks for your valuable comment. In the following, we try our best to explain our theoretical analysis on the rank of the global update more clearly.
>
> To begin with, we define the global update of FedAvg with LoRA as,
> $$
> \Delta \widetilde{\boldsymbol{W}}^T_{FedAvg} = \boldsymbol{B}_G^T\boldsymbol{A}_G^T.
> $$
>
> Here, $\boldsymbol{B}_G^T$ is a $d_1 \times r$ matrix and $\boldsymbol{A}_G^T$ is a $r \times d_2$ matrix. Based on Eq.(12), we have,
>
> $$
> rank(\Delta \widetilde{\boldsymbol{W}}^T_{FedAvg}) \leq \min (rank(\boldsymbol{B}_G^T), rank(\boldsymbol{A}_G^T))\leq r.
> $$
>
> Then, we  define the global update of FRLoRA as,
> $$
> \begin{aligned}
>     \Delta \widetilde{\boldsymbol{W}}^{T}_{FRLoRA}
>     &= \Delta\boldsymbol{W}^1 + \Delta\boldsymbol{W}^2 + \ldots + \Delta\boldsymbol{W}^T \\\\
>     &= (\boldsymbol{B}_G^1\boldsymbol{A}_G^1 - \boldsymbol{B}_G^0\boldsymbol{A}_G^0) +  (\boldsymbol{B}_G^2\boldsymbol{A}_G^2 - \boldsymbol{B}_G^0\boldsymbol{A}_G^0) + \ldots+(\boldsymbol{B}_G^T\boldsymbol{A}_G^T - \boldsymbol{B}_G^0\boldsymbol{A}_G^0)\\\\
>     &= -T\boldsymbol{B}_G^0\boldsymbol{A}_G^0 + \boldsymbol{B}_G^1\boldsymbol{A}_G^1 + \boldsymbol{B}_G^2\boldsymbol{A}_G^2 + \ldots + \boldsymbol{B}_G^T\boldsymbol{A}_G^T.
> \end{aligned}
> $$
>
> Obviously, $\Delta \widetilde{\boldsymbol{W}}^{T}_{FRLoRA}$ can be rewritten as,
>
> $$
>     \Delta \widetilde{\boldsymbol{W}}^{T}_{FRLoRA}
>     = [-T\boldsymbol{B}_G^0; \boldsymbol{B}_G^1; \boldsymbol{B}_G^2; \ldots; \boldsymbol{B}_G^T] \times [\boldsymbol{A}_G^0; \boldsymbol{A}_G^1; \boldsymbol{A}_G^2; \ldots;\boldsymbol{A}_G^T].
> $$
>
> Here, the size of $[-T\boldsymbol{B}_G^0; \boldsymbol{B}_G^1; \boldsymbol{B}_G^2; \ldots; \boldsymbol{B}_G^T]$ is $d_1 \times r(T+1)$, and the size of $[\boldsymbol{A}_G^0;\boldsymbol{A}_G^1; \boldsymbol{A}_G^2; \ldots;\boldsymbol{A}_G^T]$ is $r(T+1)  \times d_2$.
>
> Thus,  we have,
> $$
> rank(\Delta \widetilde{\boldsymbol{W}}^T_{FRLoRA}) \leq \min (r(T+1), d_1, d_2),
> $$
>
> and
> $$
> rank(\Delta \widetilde{\boldsymbol{W}}^T_{FRLoRA}) \geq rank(\Delta \widetilde{\boldsymbol{W}}^T_{FedAvg}).
> $$
>
> Finally, we note that $rank(\Delta \widetilde{\boldsymbol{W}}^T_{FRLoRA})$ is ***greater*** than $rank(\Delta \widetilde{\boldsymbol{W}}^T_{FedAvg})$,  ***only*** except that $\boldsymbol{B}_G^{0}\boldsymbol{A}_G^{0}, \boldsymbol{B}_G^1\boldsymbol{A}_G^1, \ldots, \boldsymbol{B}_G^{T}\boldsymbol{A}_G^{T}$ are ***completely linearly dependent***.
>
> We sincerely hope this theoretical analysis has effectively addressed your concerns. If you have any remaining questions or require further clarification, please do not hesitate to let us know, and we would be glad to provide further explanations.

---

> > ### Author Response · Authors · 2024-11-28
> >
> > Thanks for your valuable comment. We kindly wanted to follow up to ask if our responses have satisfactorily resolved your concerns.
> >
> > If you have any remaining questions or require further clarification, please do not hesitate to let us know, and we would be glad to provide further explanations.
> >
> > Thank you again for your efforts in reviewing our work.

---

> > > ### Comment · Reviewer_Ddf1 · 2024-11-28
> > >
> > > Thanks for your comments. But it seems that $\Delta \widetilde{\boldsymbol{W}}^T_{FedAvg}$ can also be decomposed into the sum of multiple low-rank matrices. That is, $\Delta \widetilde{\boldsymbol{W}}^T_{FedAvg} = \Delta\boldsymbol{W}^1 + \Delta\boldsymbol{W}^2 + \ldots + \Delta\boldsymbol{W}^T$, where $\Delta\boldsymbol{W}^t$ is the gradient of low-rank branch during the training at $t$ round. Then the same conclusion as $\Delta \widetilde{\boldsymbol{W}}^{T}_{FRLoRA}$  can be derived, I believe.

---

> ### Author Response · Authors · 2024-11-28
>
> Thanks for your valuable comment. We further clarify it as follows:
>
> According your description, we redefine the global update of FedAvg with LoRA as,
> $$
> \begin{aligned}
> \Delta \widetilde{\boldsymbol{W}}^T_{FedAvg} &= \boldsymbol{B}_G^T\boldsymbol{A}_G^T \\\\
> &= \boldsymbol{B}_G^0\boldsymbol{A}_G^0 + (\boldsymbol{B}_G^1\boldsymbol{A}_G^1-\boldsymbol{B}_G^0\boldsymbol{A}_G^0) +
> (\boldsymbol{B}_G^2\boldsymbol{A}_G^2-\boldsymbol{B}_G^1\boldsymbol{A}_G^1) + \ldots + (\boldsymbol{B}_G^T\boldsymbol{A}_G^T-\boldsymbol{B}_G^{T-1}\boldsymbol{A}_G^{T-1}) \\\\
> &= \boldsymbol{B}_G^0\boldsymbol{A}_G^0 + \Delta \boldsymbol{W}^1 + \Delta \boldsymbol{W}^2 + \ldots + \boldsymbol{W}^T.
> \end{aligned}
> $$
>
> We can observe that $\Delta \boldsymbol{W}^{t} = \boldsymbol{B}_G^t\boldsymbol{A}_G^t - \boldsymbol{B}_G^{t-1}\boldsymbol{A}_G^{t-1}$ and $\Delta \boldsymbol{W}^{t-1} = \boldsymbol{B}_G^{t-1}\boldsymbol{A}_G^{t-1}-\boldsymbol{B}_G^{t-2}\boldsymbol{A}_G^{t-2}$ are **NOT** independent. We note that FedAvg only updates the low-rank matrices and uses them as the initialization for the next training round. Consequently, these terms can still be ***merged*** into a low rank matrix, $\boldsymbol{B}_G^0\boldsymbol{A}_G^0 + \Delta \boldsymbol{W}^1 + \Delta \boldsymbol{W}^2 + \ldots + \boldsymbol{W}^T = \boldsymbol{B}_G^T\boldsymbol{A}_G^T$. Here, $\boldsymbol{B}_G^T$ is a $d_1 \times r$ matrix and $\boldsymbol{A}_G^T$ is a $r \times d_2$ matirx.
>
> Thus, we have,
>
> $$
> rank(\Delta \widetilde{\boldsymbol{W}}^T_{FedAvg}) \leq \min(rank(\boldsymbol{B}_G^T), rank(\boldsymbol{A}_G^T)) \leq r.
> $$
>
> In contrast, the global update of FRLoRA can be expressed as,
> $$
> \begin{aligned}
>     \Delta \widetilde{\boldsymbol{W}}^{T}_{FRLoRA}
>     &= \Delta\boldsymbol{W}^1 + \Delta\boldsymbol{W}^2 + \ldots + \Delta\boldsymbol{W}^T \\\\
>     &= (\boldsymbol{B}_G^1\boldsymbol{A}_G^1 - \boldsymbol{B}_G^0\boldsymbol{A}_G^0) +  (\boldsymbol{B}_G^2\boldsymbol{A}_G^2 - \boldsymbol{B}_G^0\boldsymbol{A}_G^0) + \ldots+(\boldsymbol{B}_G^T\boldsymbol{A}_G^T - \boldsymbol{B}_G^0\boldsymbol{A}_G^0)\\\\
>     &= -T\boldsymbol{B}_G^0\boldsymbol{A}_G^0 + \boldsymbol{B}_G^1\boldsymbol{A}_G^1 + \boldsymbol{B}_G^2\boldsymbol{A}_G^2 + \ldots + \boldsymbol{B}_G^T\boldsymbol{A}_G^T.
> \end{aligned}
> $$
>
> Given that FRLoRA ***reinitializes*** the low-rank matrices in each training round and ***directly updates*** the global model using the updated global low-rank matrices, $\Delta \boldsymbol{W}^{t} = \boldsymbol{B}_G^t\boldsymbol{A}_G^t - \boldsymbol{B}_G^{0}\boldsymbol{A}_G^{0}$ and $\Delta \boldsymbol{W}^{t-1} = \boldsymbol{B}_G^{t-1}\boldsymbol{A}_G^{t-1}-\boldsymbol{B}_G^{0}\boldsymbol{A}_G^{0}$ are ***independent*** and can **NOT** be merged.
>
> And $\Delta \widetilde{\boldsymbol{W}}^{T}_{FRLoRA}$ can be rewritten as,
>
> $$
>     \Delta \widetilde{\boldsymbol{W}}^{T}_{FRLoRA}
>     = [-T\boldsymbol{B}_G^0; \boldsymbol{B}_G^1; \boldsymbol{B}_G^2; \ldots; \boldsymbol{B}_G^T] \times [\boldsymbol{A}_G^0; \boldsymbol{A}_G^1; \boldsymbol{A}_G^2; \ldots;\boldsymbol{A}_G^T].
> $$
>
> Here, the size of $[-T\boldsymbol{B}_G^0; \boldsymbol{B}_G^1; \boldsymbol{B}_G^2; \ldots; \boldsymbol{B}_G^T]$  is $d_1 \times r(T+1)$, and the size of $[\boldsymbol{A}_G^0;\boldsymbol{A}_G^1; \boldsymbol{A}_G^2; \ldots;\boldsymbol{A}_G^T]$  is $r(T+1)  \times d_2$.
>
>
> Compared to FedAvg, FRLoRA ***extends*** the global update from two low-rank matrices, $\boldsymbol{B}_G^T$ and $\boldsymbol{A}_G^T$, to two high-rank matrices, $[-T\boldsymbol{B}_G^0; \boldsymbol{B}_G^1; \boldsymbol{B}_G^2; \ldots; \boldsymbol{B}_G^T]$ and $[\boldsymbol{A}_G^0;\boldsymbol{A}_G^1; \boldsymbol{A}_G^2; \ldots;\boldsymbol{A}_G^T]$.
>
>
>
> Thus,  we have,
> $$
> rank(\Delta \widetilde{\boldsymbol{W}}^T_{FRLoRA}) \leq \min (r(T+1), d_1, d_2),
> $$
> and
> $$
> rank(\Delta \widetilde{\boldsymbol{W}}^T_{FRLoRA}) \geq rank(\Delta \widetilde{\boldsymbol{W}}^T_{FedAvg}).
> $$
>
> Finally, we note that $rank(\Delta \widetilde{\boldsymbol{W}}^T_{FRLoRA})$ is ***greater*** than $rank(\Delta \widetilde{\boldsymbol{W}}^T_{FedAvg})$,  ***only*** except that $\boldsymbol{B}_G^{0}\boldsymbol{A}_G^{0}, \boldsymbol{B}_G^1\boldsymbol{A}_G^1, \ldots, \boldsymbol{B}_G^{T}\boldsymbol{A}_G^{T}$ are ***completely linearly dependent***.
>
> Besides, we also recorded the rank of the global update for FedAvg (r=16) after training for 100 rounds on MetaMathQA. The ***average*** rank of $\Delta \widetilde{\boldsymbol{W}}^T_{FedAvg}$  at each layer is **16**. In contrast, the ***average*** rank of $\Delta\widetilde{\boldsymbol{W}}^{T}_{FRLoRA}$ at each layer for FRLoRA (r=16) reached **82**.  This further proves that FRLoRA can achieve the global update with a higher rank.
>
> We sincerely hope our response has effectively addressed your concerns. If you have any remaining questions or require further clarification, please do not hesitate to let us know, and we would be glad to provide further explanations.

---

> ### Comment · Reviewer_Ddf1 · 2024-11-29
>
> Thanks for your explanation.  My concerns have been addressed and thereby I raising my score. Hope you can include these clarifications in your future mannuscript.

---

> > ### Author Response · Authors · 2024-11-29
> >
> > Thanks for your valuable time to respond to our feedback！
> >
> > We are very happy to see that your concerns have been fully addressed :)
> >
> > We will include these clarification in our final mannuscript.  Thank you once again for reviewing our work.

---

### Official Review · Reviewer_Acqq · 2024-11-02

**Soundness:** 3
**Presentation:** 2
**Contribution:** 3
**Rating:** 8
**Confidence:** 5

**Summary:**

This paper explores the challenges of using LoRA in Federated Learning (FL) for fine-tuning large language models (LLMs) on non-IID data. The authors identify that LoRA in FL struggles to learn global knowledge due to two key issues: extrinsic client drift and an intrinsic constrained update space. The authors propose FRLoRA (Federated Residual Low-Rank Adaptation) to overcome these challenges. FRLoRA initializes the LoRA matrices  $A$  and  $B$  using the Singular Value Decomposition (SVD) of the initial weights  $W_0$, while freezing the residual initial weight. During training, the locally updated LoRA matrices are aggregated and combined with the residual weight, creating the initial model for the next training round. This method addresses both client drift and limited update space, of which theoretical justification is presented. The experiments on nine language model benchmarks show that FRLoRA consistently outperforms various baseline methods.

**Strengths:**

**Addressing Non-IID Challenges in FL with LoRA:** FRLoRA tackles an important challenge in federated fine-tuning of LLMs by addressing both client drift and constrained parameter space through aggregated $\Delta W$ updates via SVD.

**Theoretical Justification:** The authors provide a simple theoretical justification of FRLoRA using a rank analysis of the parameter space, although the rank analysis can be far from the practice.

**Comprehensive Evaluation:** The study provides extensive experiments across nine benchmarks, covering Natural Language Understanding and Generation, validating FRLoRA’s efficacy in terms of performance and communication cost.

**Weaknesses:**

**Insufficient Evidence on the Problem Formulation:** The authors characterize the client drift using Figure 1 (c-d) but it is not convincing. To be specific, the norm of $\Delta W$ and the standard deviation can be simply reduced by the other factors, e.g., learning rate. Perhaps, it would be better to (i) focus on a single round rather than the sequence of rounds and (ii) observe the distribution of cosine similarity rather than the standard deviation. It is also helpful to analyze the other FL methods.

In addition, the authors claim that the constrained parameter space is one of the major problems in the naive method (FedAVG + LoRA). The simple rank analysis supports this claim. However, it is not thoroughly studied in empirical analysis. Hypothetically, the constrained parameter space problem can be simply addressed by employing large rank $r$.



**Limitations in Non-IID Testing Scenarios:** Experiments use a Dirichlet (0.5) Non-IID setting with five clients for binary classification, while NLG tests are performed on IID data. Broader Non-IID experiments would better showcase FRLoRA's robustness instead of just increasing the number of benchmarks.


**Limited Comparision to Existing Works:** The baselines used for comparison, such as FedYogi, FedAdam, and FedProx, do not integrate LoRA, making comparisons with these methods less relevant. Only FFA-LoRA offers a directly comparable baseline. Apparently, there is another LoRA method for FL, FlexLoRA (https://arxiv.org/abs/2402.11505, Feb 2024), in which a similar idea of using SVD per round is proposed. In addition, the idea of using the residual is analog to the one in Chain of LoRA (https://arxiv.org/abs/2401.04151, Jan 2024), which needs to be discussed.

**Questions:**

## Questions
**Does FRLoRA Fully Address the Intrinsic Challenge?**: While FRLoRA expands the parameter space globally, it does not necessarily address the constrained optimization space at the local level. Can the expanded global space fully mitigate intrinsic issues without modifying local optimization constraints?

**Catastrophic Forgetting at Low Ranks**: Direct updates to $W_0$ in a low-rank context may risk catastrophic forgetting. Can FRLoRA’s SVD initialization mitigate this risk, or could additional ablation studies on learning rates clarify this?

**Comparison to Chain of LoRA’s Residual Framework**: Both FRLoRA and Chain of LoRA use residual structures for fine-tuning. How does FRLoRA’s SVD-based initialization impact its effectiveness relative to the iterative approach of Chain of LoRA? Further analysis of residual efficacy could enhance the comparative evaluation.

**More Diverse Data Heterogeneity**: Is there an empirical study for NLG with severe heterogeneity?

---

> ### Author Response · Authors · 2024-11-23
> **Official Comment by Authors （1/3）**
>
> We thank the reviewer for his/her constructive comments and provide our point-wise replies as follows.
>
> > **Q1:** The distribution of cosine similarity in a single round.
>
> Thanks for your suggestion. We analyzed the cosine similarity of local updates, $\Delta\boldsymbol{W}_k^t$, across different clients in a single round for FedAvg (non-IID), FedAvg (IID), and FRLoRA (non-IID). All methods are trained for the same number of rounds, and the results have been included in the revision (Appendix C.5 Figure 3).
>
> The results show that the cosine similarity between clients in FedAvg (IID) (see Figure 3 (b)) is distributed significantly higher than in FedAvg (non-IID) (see Figure 3 (a)), and FRLoRA (non-IID) (see Figure 3 (c)) exhibits a ***higher*** distribution compared to FedAvg (non-IID). This further demonstrates the effectiveness of FRLoRA in ***reducing*** client drift, promoting more ***consistent*** model convergence under data heterogeneity.
>
> > **Q2:** Empirical analysis of the constrained parameter space.
>
> We have conducted this empirical analysis in our submitted paper (Table 10). And we present these results below.
>
> | Method       | GSM8K (r=16) | Math (r=16) | Avg. (r=16) | GSM8K (r=32) | Math (r=32) | Avg. (r=32) | GSM8K (r=64) | Math (r=64) | Avg. (r=64) |
> |--------------|--------------|-------------|-------------|--------------|-------------|-------------|--------------|-------------|-------------|
> | FedAvg       | 32.67        | 4.64        | 18.65       | 34.95        | 4.48        | 19.71       | 37.45        | 5.38        | 21.41       |
> | FedProx      | 32.29        | 4.32        | 18.30       | 35.40        | 4.66        | 20.03       | 36.39        | 4.98        | 20.68       |
> | SCAFFOLD     | 32.97        | 4.70        | 18.84       | 35.78        | 5.08        | 20.43       | 32.37        | 4.64        | 18.50       |
> | FedAvgM      | 32.44        | 4.42        | 18.43       | 34.79        | 4.64        | 19.71       | 35.57        | 4.72        | 20.14       |
> | FedAdagrad   | 28.65        | 4.18        | 16.41       | 29.64        | 4.06        | 16.85       | 31.76        | 4.46        | 18.31       |
> | FedYogi      | 30.32        | 4.00        | 17.16       | 30.09        | 4.04        | 17.06       | 33.96        | 4.40        | 19.33       |
> | FedAdam      | 31.23        | 4.14        | 17.68       | 31.84        | 4.12        | 17.98       | 34.26        | 5.16        | 39.44       |
> | FFA-LoRA     | 25.17        | 3.60        | 14.38       | 28.05        | 3.78        | 15.91       | 31.00        | 4.50        | 17.75       |
> | **FRLoRA (Ours)** | **39.57**    | **5.60**    | **22.58**    | **44.27**    | **5.22**    | **24.74**    | **45.56**    | **6.88**    | **26.22**    |
>
> (1) As shown, increasing the rank $r$ in FedAvg + LoRA does alleviate the constrained parameter space issue, leading to performance improvements. However, this improvement incurs ***larger*** communication overhead.
>
> (2) Unlike simply increasing $r$, FRLoRA effectively performs global updates in a higher-rank space with the same $r$ as FedAvg. It can be observed that FRLoRA ($r$ = 16) achieved even ***higher*** performance than FedAvg (r=64), strongly demonstrating our claim. We have clarified this in the revision (Lines 1073-1074).
>
>
>
> > **Q3:** Limited non-IID scenarios.
>
> We clarify it as follows:
>
> (1) We would like to clarify that the datasets used in our experiments, i.e., FedAya, Fed-ChatbotIT, and Fed-WildChat, are **INDEED** real-world datasets ***with data heterogeneity*** and suggested by FedLLM-Bench [1]. The results (Tables 3 and 4) strongly demonstrate the effectiveness of our method for NLG tasks under severe heterogeneity.
>
> (2) We further investigate the impact of data heterogeneity on our method for NLU tasks. The results have been included in the revision (Appendix C.6 Figure 4), indicating that the accuracy of all methods increases with the increase of $\beta$, and FRLoRA ***significantly outperforms*** the other methods at different values of $\beta$. Moreover, the improvement of FRLoRA over other methods is ***larger*** when $\beta$ is small, indicating the effectiveness of FRLoRA for NLU tasks under data heterogeneity.

---

> ### Author Response · Authors · 2024-11-23
> **Official Comment by Authors （2/3）**
>
> > **Q4:** Add comparison with FlexLoRA.
>
> Thanks for your suggestion. We have compared our method against FlexLoRA in the revision (Tables 1, 2, 3, and 4). The results, as presented below, show that our method consistently **outperforms** FlexLoRA across nine different benchmarks including both NLG amd NLU tasks.
>
>
> *Table 1: NLU Tasks*
> | Method      | RTE   | COLA  | 20NG  | QNLI   |
> |-------------|-------|-------|-------|--------|
> | FlexLoRA    | 70.28 | 62.56 | 65.98 | 90.03  |
> | **FRLoRA (Ours)** | **75.81** | **64.80** | **69.41** | **91.10** |
>
>
> *Table 2: NLG Tasks with MetaMathQA and Alpaca-GPT4*
> | Method       | MetaMathQA (GSM8K) | MetaMathQA (Math) | MetaMathQA (Avg.) | Alpaca-GPT4 (Vicuna) | Alpaca-GPT4 (MT-1) | Alpaca-GPT4 (MT-2) | Alpaca-GPT4 (MT-Avg.) | Alpaca-GPT4 (Avg.) |
> |--------------|--------------------|-------------------|-------------------|----------------------|--------------------|--------------------|-----------------------|--------------------|
> | FlexLoRA     | 34.09             | 4.31             | 19.20            | 7.884               | 4.561             | 2.012             | 3.286                | 4.435             |
> | **FRLoRA (Ours)** | **44.27**       | **5.22**         | **24.74**        | **8.044**           | **4.775**         | **2.481**         | **3.635**            | **4.733**         |
>
>
> *Table 3: Real-world NLG Tasks with Fed-Aya*
> | Method       | ar   | en   | es   | fr   | pt   | ru   | te   | zh   | Avg. |
> |--------------|------|------|------|------|------|------|------|------|------|
> | FlexLoRA     | 2.60 | 8.20 | **6.25** | 5.05 | 4.70 | 5.20 | **1.85** | 4.75 | 4.70 |
> | **FRLoRA (Ours)** | **4.45** | 7.75 | 6.15 | **6.65** | **4.75** | **6.25** | 1.55 | **6.95** | **5.56** |
>
> *Table 4: Real-world NLG Tasks with Fed-ChatbotIT and Fed-WildChat*
> | Method       | Fed-ChatbotIT (MT-1) | Fed-ChatbotIT (Vicuna) | Fed-ChatbotIT (Ref-GPT4) | Fed-ChatbotIT (Avg.) | Fed-WildChat (MT-1) | Fed-WildChat (Vicuna) | Fed-WildChat  (Ref-GPT4) | Fed-WildChat (Avg.) |
> |--------------|--------------------|-------------------|-------------------|----------------------|--------------------|--------------------|-----------------------|--------------------|
> | FlexLoRA     | 4.17| 7.02| 5.40| 5.53| 4.88| 7.91 |5.78| 6.19  |
> | **FRLoRA (Ours)** | **4.31**       | **7.49**         | **5.62**        | **5.80**           | **4.64**         | **8.24**         | **7.00**            | **6.63**         |
>
>
> > **Q5:** Does FRLoRA fully address the intrinsic challenge?
>
> We answer this question as follows:
>
> (1) We agree that FRLoRA does not fully address the constrained optimization space at the local level. In this work, FRLoRA ***mainly*** focuses on addressing the intrinsic challenges from the ***global perspective***. It expands the parameter space of global updates to a higher-rank space, allowing for a ***better*** representation of the diverse knowledge learned from different clients. Extensive experiments demonstrate its effectiveness.
>
> (2) Since the global model is shared by all clients, we believe that local optimization can also ***benefit*** from the global expansion of the parameter space. We will explore this aspect in our future work.

---

> ### Author Response · Authors · 2024-11-23
> **Official Comment by Authors （3/3）**
>
> > **Q6:** Catastrophic forgetting at low ranks.
>
> To ***directly measure*** catastrophic forgetting at low ranks, we conducted an exploratory experiment, i.e., fine-tuned CLIP/ViT-B32 on CIFAR-10 using FedAvg and FRLoRA. Specifically, the CIFAR-10 training set was divided into 10 clients based on a Dirichlet distribution with $\beta = 0.5$. The model was optimized using AdamW with a learning rate of 5e-5 and a batch size of 64. The fine-tuned models were evaluated on the CIFAR-100 test set in a zero-shot manner, and their performance on the CIFAR-10 test set was also reported. To measure the fine-tuning performance and catastrophic forgetting, we also reported the zero-shot accuracy of the original CLIP/ViT-B32 on the test set of CIFAR-10 and CIFAR-100. The results are presented below:
>
> |Method | CIFAR-10| CIFAR-100 |
> |:----:|:----:|:----:|
> | CLIP/ViT-B32 (Zero-shot) | 84.75 | 56.07 |
> |FedAvg| 87.49 | 38.88 |
> |FRLoRA| 94.72 | 40.32 |
>
>
> It can be observed that fine-tuning LoRA on downstream tasks leads to forgetting. However, compared to FedAvg, FRLoRA achieves ***better*** zero-shot accuracy on CIFAR-100, FRLoRA (40.32) vs. FedAvg (38.88). This indicates that FRLoRA's initialization approach ***can partially mitigate*** the forgetting issue, as it ensures that local training is consistently performed in the principal singular space of the pre-trained model.
> Furthermore, this work primarily aims to achieve effective federated fine-tuning ***rather than*** addressing forgetting. Our focus lies in improving model performance on ***downstream tasks***, thereby facilitating the ***practical application*** of LLMs in real-world scenarios. The extensive experimental results presented in this paper, including the findings on CIFAR-10, provide strong evidence that our method successfully ***reaches*** this objective.
>
>
>
>  > **Q7:** Discussion and comparison with Chain of LoRA’s （COLA） residual framework.
>
> (1) Although FRLoRA and COLA have similar ideas, there exists ***differences*** in their residual framework. COLA ***directly merges*** the updated low-rank matrices $\boldsymbol{B}_G^t\boldsymbol{A}_G^t$ back into the model in each round, whereas FRLoRA ***only uses*** the residual updates $\boldsymbol{B}_G^t\boldsymbol{A}_G^t - \boldsymbol{B}_G^0\boldsymbol{A}_G^0$ to update the parameters of the model.
>
> (2) If we apply the residual framework of CoLA in FL with FRLoRA’s SVD-based initialization, then Eq.(9) should be rewritten as: $\Delta\boldsymbol{W}^{t} = \boldsymbol{B}_G^t\boldsymbol{A}_G^t$
>
> And after global update, we need to decompose the new weights of global model into a low-rank structure using Eq.(5)-(7), and use them to initialize the low-rank matrices. Acually, it is FRLoRA-v3, a variant in our ablation study (Section 4.4). We present the results as follows.
>
> | Method       | NLU (RTE) | NLU (20NG) | MetaMathQA (GSM8K) | MetaMathQA (Math) | MetaMathQA (Avg.) | Fed-WildChat (MT-1) | Fed-WildChat (Vicuna) | Fed-WildChat (Ref-GPT4) | Fed-WildChat (Avg.) |
> |--------------|---------|-----------|------------------|-----------------|-----------------|-------------------|---------------------|-----------------------|-------------------|
> | FRLoRA + COLA   | 58.62| 63.19| 39.08| 5.01| 22.04| 4.43| 8.03| 6.34 | 6.26 |
> | **FRLoRA (Ours)** | **75.81** | **69.41** | **44.27**      | **5.22**       | **24.74**       | **4.64**          | **8.24**            | **7.00**              | **6.63**         |
>
> We can observe that its results are ***significantly worse*** than ours, primarily due to the frequent SVD decompositions causing oscillations in the parameter space, which leads to unstable convergence. Additionally, it also incurs ***extra training time*** for large models like LLaMA-2-7B, as shown below.
>
>
> |Model | Time (s) |
> |:----:|:----:|
> |LLaMA-2-7B| 197.82 $×$ T |
>
>
>
> **References**
>
> [1] Ye, Rui, et al. "FedLLM-Bench: Realistic Benchmarks for Federated Learning of Large Language Models." arXiv preprint arXiv:2406.04845 (2024).

---

> > ### Comment · Reviewer_Acqq · 2024-11-25
> > **Thanks for the further analysis**
> >
> > Thanks for the further analysis. As most of my concerns have addressed, I increase my score from 6 to 7.

---

> > > ### Author Response · Authors · 2024-11-25
> > >
> > > Thanks for your valuable time to respond to our feedback！
> > >
> > > We are very happy to see that your concerns have been fully addressed :)

---

### Official Review · Reviewer_vtoo · 2024-11-04

**Soundness:** 3
**Presentation:** 3
**Contribution:** 2
**Rating:** 6
**Confidence:** 2

**Summary:**

The paper presents Federated Residual Low-Rank Adaptation (FRLoRA), a federated learning method that improves LoRA for LLMs. FRLoRA overcomes challenges like constrained parameter spaces and client drift by updating in a higher-rank parameter space and reinitializing local low-rank matrices based on the principal singular values of pre-trained weights. Experiments show that FRLoRA consistently outperforms existing federated learning methods across benchmarks in NLU and NLG tasks.

**Strengths:**

1. The paper introduces a unique adaptation of LoRA for federated learning with residual low-rank updates and reinitialization in the principal singular space.

2. Extensive experiments on multiple benchmarks confirm FRLoRA’s robust improvements across NLU and NLG tasks in various federated settings.

**Weaknesses:**

1. The reinitialization of local low-rank matrices and the use of SVD may introduce significant computational overhead, particularly for large-scale models. An in-depth analysis of this aspect or potential optimizations would provide added value to the paper.

2. Although the method shows performance improvements overall, in Table 3, the proposed approach does not appear to dominate across all evaluations. A similar pattern is observed in Table 4, indicating that while the method is strong, it may not consistently outperform existing baselines in certain subcases.

3. While the paper briefly touches on SVD stability, it lacks a detailed analysis of potential impacts and does not explore alternative methods for scenarios where SVD might lead to convergence issues. Additional insights on this aspect could reinforce the robustness and general applicability of the approach.

**Questions:**

Q1: Could the authors provide further analysis on the computational impact of using SVD for reinitialization in terms of time and memory requirements?

Q2: Could the authors provide further analysis on the behaviour on Table 3 and 4?

Q2:How does FRLoRA handle scenarios where singular value decomposition may be unstable? Are there alternative initialization strategies?

---

> ### Author Response · Authors · 2024-11-23
> **Official Comment by Authors**
>
> We thank the reviewer for his/her constructive comments and provide our point-wise replies as follows.
>
> > **Q1:** Time and memory of SVD computation.
>
> We have conducted an analysis of the time and memory requirements for the SVD computation. The results are summarized below:
>
> |Model | Time (s) | Peak Memory (MB) |
> |:----:|:----:|:----:|
> |RoBERTa-base| 1.48| 0.25  |
> |LLaMA-2-7B| 197.82 | 0.65  |
>
> As shown, the computational cost of SVD is ***minimal*** in terms of memory, with peak memory consumption remaining ***under 1 MB*** for both models. While the time cost scales with model size, SVD is performed only **ONCE**, rendering its overall impact ***negligible*** compared to the whole training phase. The results have been included in the revision (Appendix C.4 Table 15).
>
>
>
> > **Q2:** Further analysis on Tables 3 and 4.
>
> Unlike NLU tasks, NLG tasks involve ***different*** datasets for training and testing.  For instance, in Table 4, the model is trained on Fed-chatbotIT but evaluated on three benchmarks: MT-Bench, Vicuna, and Ref-GPT4. Due to variations in data distributions or task-specific characteristics across benchmarks, certain subtask metrics may not achieve the best performance. However, the **GOAL** of fine-tuned LLMs is to achieve the ***generalized*** performance across diverse downstream  tasks. Consequently, ***average*** task performance holds ***greater*** importance than local advantages in individual subtasks, especially considering that real-world data distributions are typically heterogeneous and uncontrollable. FRLoRA's performance aligns **MORE** closely with these practical requirements.
>
> > **Q3:** Discussion about SVD.
>
>
> We answer your question as follows:
>
> (1) In FRLoRA, SVD is employed **ONLY** during the initialization phase to decompose the pre-trained weights into a low-rank structure. In our experiments across **NINE** different benchmarks, SVD has been empirically proven to be ***stable*** when applied to these weight matrices, as they are generally well-conditioned. Additionally, SVD is performed only **ONCE** during the initialization, which ***minimizes*** the risk of instability.
>
> (2) For extreme cases, such as with ill-conditioned or noisy weight matrices, SVD may become unstable. Although such scenarios are **RARE**, we can replace standard SVD with randomized SVD or matrix sketching techniques, which provide more stable decompositions in challenging conditions. Furthermore, incorporating regularization methods like Tikhonov regularization or performing truncated SVD can help stabilize the decomposition process when facing instability. These alternative strategies offer flexibility in handling unstable SVD scenarios, ensuring robust and stable initialization in extreme cases.

---

> ### Comment · Reviewer_vtoo · 2024-11-27
>
> I appreciate the dedicated work that the author invested in the rebuttal. My concerns have been addressed, and I will keep my positive rating.

---

> > ### Author Response · Authors · 2024-11-28
> >
> > Thanks for your valuable time to respond to our feedback！
> >
> > We are very happy to see that your concerns have been fully addressed :)

---

### Official Review · Reviewer_pHac · 2024-11-05

**Soundness:** 3
**Presentation:** 3
**Contribution:** 3
**Rating:** 6
**Confidence:** 4

**Summary:**

This paper introduces a method known as FRLoRA (Federated Residual Low-Rank Adaption) to address the problem of Parameter-Efficient Fine-Tuning (PEFT) for Large Language Models (LLMs) in Federated Learning (FL) scenarios. FRLoRA updates global model parameters by introducing a residual low-rank matrix product and reinitializes local low-rank matrices in each training round to mitigate client drift. The paper compares FRLoRA with existing FL baseline methods across multiple datasets in natural language understanding and generation tasks, demonstrating its consistent superiority and validating the effectiveness of the proposed FRLoRA.

**Strengths:**

1. FRLoRA effectively addresses the significant impact of data heterogeneity on PEFT in FL scenarios by accumulating the product of residual low-rank matrices, enabling the global model to learn more comprehensive knowledge.
2. By reinitializing local low-rank matrices at each training round, FRLoRA alleviates client drift issues, enhancing the convergence and performance of the model.
3. Extensive experiments validate the performance improvements of FRLoRA over existing methods across various natural language processing tasks.

**Weaknesses:**

1. The updates derived from averaging the uploaded matrices A and B at the server differ from the averaged updates across clients, and this discrepancy is likely to amplify in heterogeneous data scenarios. [Improving LoRA in Privacy-preserving Federated Learning]
2. Although the paper proposes extending the parameter space through residual updates, it does not clearly specify whether the rank of the model parameters is strictly improved after each residual update, lacking further explanation on how this mechanism operates, particularly regarding its impact on model rank and representational capacity.

**Questions:**

See Weaknesses

---

> ### Author Response · Authors · 2024-11-23
> **Official Comment by Authors**
>
> We thank the reviewer for his/her constructive comments and provide our point-wise replies as follows.
>
> > **Q1:** Discrepancy between the uploaded matrices and the averaged updates.
>
> We have conducted an exploratory experiment by fixing the $\boldsymbol{A}$ matrix in FRLoRA, with the results presented below.
>
> | Method       | NLU (RTE) | NLU (20NG) | MetaMathQA (GSM8K) | MetaMathQA (Math) | MetaMathQA (Avg.) | Fed-WildChat (MT-1) | Fed-WildChat (Vicuna) | Fed-WildChat (Ref-GPT4) | Fed-WildChat (Avg.) |
> |--------------|---------|-----------|------------------|-----------------|-----------------|-------------------|---------------------|-----------------------|-------------------|
> | FFA-LoRA| 68.69| 66.88| 28.05 | 3.78| 15.91|4.81| 7.99| 5.88| 6.22|
> | FRLoRA + FFA-LoRA   | 72.06 | 67.58 | 34.85 | 4.79 | 19.82 | 4.52 | 7.87 | 5.98 | 6.12 |
> | **FRLoRA (Ours)** | **75.81** | **69.41** | **44.27**      | **5.22**       | **24.74**       | **4.64**          | **8.24**            | **7.00**              | **6.63**         |
>
> While this variant outperforms FFA-LoRA, its performance is ***inferior*** to ours,  as fixing $\boldsymbol{A}$ greatly **restricts** the learning capacity. This finding shows that while averaging the uploaded matrices A and B at the server differs from the averaged updates across clients, it can still serve as a ***suitable tradeoff*** between discrepancy and performance in contrast to FFA-LoRA.
>
> > **Q2:** Further explanation on the residual update mechanism.
>
> We explain this from the following two aspects:
>
> (1) In our method, the rank of the model parameters is **NOT** strictly improved after each residual update. ***Instead***, the iterative accumulative effect of these residual updates (Eq.(13)), when applied to the global model’s parameters, expands the parameter space of global updates to a higher-rank space. This mechanism inherently enhances the model’s representational capacity, enabling a ***better*** representation of the diverse knowledge learned from different clients.
>
> (2) Besides, we have provided results with different values of $r$ in our submitted paper (Table 10). The results are presented below:
>
> | Method       | GSM8K (r=16) | Math (r=16) | Avg. (r=16) | GSM8K (r=32) | Math (r=32) | Avg. (r=32) | GSM8K (r=64) | Math (r=64) | Avg. (r=64) |
> |--------------|--------------|-------------|-------------|--------------|-------------|-------------|--------------|-------------|-------------|
> | FedAvg       | 32.67        | 4.64        | 18.65       | 34.95        | 4.48        | 19.71       | 37.45        | 5.38        | 21.41       |
> | FedProx      | 32.29        | 4.32        | 18.30       | 35.40        | 4.66        | 20.03       | 36.39        | 4.98        | 20.68       |
> | SCAFFOLD     | 32.97        | 4.70        | 18.84       | 35.78        | 5.08        | 20.43       | 32.37        | 4.64        | 18.50       |
> | FedAvgM      | 32.44        | 4.42        | 18.43       | 34.79        | 4.64        | 19.71       | 35.57        | 4.72        | 20.14       |
> | FedAdagrad   | 28.65        | 4.18        | 16.41       | 29.64        | 4.06        | 16.85       | 31.76        | 4.46        | 18.31       |
> | FedYogi      | 30.32        | 4.00        | 17.16       | 30.09        | 4.04        | 17.06       | 33.96        | 4.40        | 19.33       |
> | FedAdam      | 31.23        | 4.14        | 17.68       | 31.84        | 4.12        | 17.98       | 34.26        | 5.16        | 39.44       |
> | FFA-LoRA     | 25.17        | 3.60        | 14.38       | 28.05        | 3.78        | 15.91       | 31.00        | 4.50        | 17.75       |
> | **FRLoRA (Ours)** | **39.57**    | **5.60**    | **22.58**    | **44.27**    | **5.22**    | **24.74**    | **45.56**    | **6.88**    | **26.22**    |
>
> As shown, FRLoRA ($r$ = 16) outperforms FedAvg ($r$=16) and even achieves ***higher*** performance than FedAvg (r=64). Besides, we observed that when training FRLoRA ($r=16$) on MetaMathQA after 100 rounds, the ***average*** rank of residual accumulation at each layer reached **82**. These results ***strongly*** demonstrate the effectiveness of our method in expanding the parameter space of global updates.

---

> > ### Author Response · Authors · 2024-11-25
> > **Look forward to your post-rebuttal feedback**
> >
> > As the discussion period draws to a close soon, we extend our sincere gratitude to you for the valuable time and insightful comments.
> >
> > In our previous response, we have carefully studied your comments and made detailed responses summarized below:
> >
> > 1. Conducted an empirical analysis of the discrepancy between the uploaded matrices and the averaged updates.
> > 2. Provided further explanation on the residual update mechanism.
> >
> > We sincerely hope our responses have effectively addressed your concerns. If you have any remaining questions or require further clarification, please do not hesitate to let us know, and we would be glad to provide further explanations
> >
> > Thank you again for your efforts in reviewing our work.

---

> > > ### Author Response · Authors · 2024-11-29
> > >
> > > Thanks for your valuable comment. We kindly wanted to follow up to ask if our responses have satisfactorily resolved your concerns.
> > >
> > > If you have any remaining questions or require further clarification, please do not hesitate to let us know, and we would be glad to provide further explanations.
> > >
> > > Thank you again for your efforts in reviewing our work.

---

> > > > ### Comment · Reviewer_pHac · 2024-11-29
> > > >
> > > > Thank you for your response. My concerns have been addressed and I raised my rating.

---

> > > > > ### Author Response · Authors · 2024-11-29
> > > > >
> > > > > Thanks for your valuable time to respond to our feedback！
> > > > >
> > > > > We are very happy to see that your concerns have been fully addressed :)

---

### Official Review · Reviewer_RHQ7 · 2024-11-10

**Soundness:** 3
**Presentation:** 3
**Contribution:** 2
**Rating:** 5
**Confidence:** 3

**Summary:**

This paper introduces FRLoRA, a novel Federated Learning (FL) method that addresses limitations in global knowledge learning by combining residual low-rank adaptation with periodic recalibration of local models. FRLoRA expands the effective parameter space during global updates and mitigates client drift by reinitializing local model components with principal components from the pre-trained model.  Extensive experiments across various natural language processing tasks demonstrate FRLoRA's superior performance compared to existing FL methods.

**Strengths:**

The paper is clearly presented. The proposed method is sound. Experiment results are comprehensive and validate the quality improvement of the proposed method.

**Weaknesses:**

This paper presents a novel Federated Learning method, FRLoRA, which aims to improve global knowledge learning. However, the paper's practical value is limited due to two main weaknesses:

Lack of a compelling use case: The authors fail to clearly motivate the need for their proposed method within a real-world application scenario. While the technical contributions may be sound, the lack of a clear practical context diminishes the paper's impact and relevance.

High communication cost: Despite claims of efficiency, the method introduces significant communication overhead, hindering its practicality in real-world deployments, where many LoRA customers care about.

**Questions:**

Can the authors should further explore and demonstrate the method's effectiveness in a specific application domain and address the communication cost limitations to enhance its practical value.

---

> ### Author Response · Authors · 2024-11-23
> **Official Comment by Authors**
>
> We thank the reviewer for his/her constructive comments and provide our point-wise replies as follows.
>
> > **Q1:** Lack of a compelling real-world use case.
>
> We would like to clarify that the datasets used in our experiments, i.e., FedAya, Fed-ChatbotIT, and Fed-WildChat, are **INDEED** derived from real-world scenarios and suggested by FedLLM-Bench [1]. Their results in Tables 3 and 4 demonstrate consistently strong performance of FRLoRA across different real-world NLG tasks, providing both a compelling use case and solid empirical evidence of our method's effectiveness in real-world application scenarios.
>
>
> > **Q2:** High communication cost.
>
> The communication cost analysis presented in the submitted paper only shows the total communication cost (**Cost.ToTal**) in *Setting 2*. This may lack key information and be misleading about the communication overhead of our method. In the revision (Appendix C.3 Table 14), we have provided a more detailed communication cost analysis, as shown below.
>
> | **Method**                  | **Cost.Active (MB)**     | **Cost.Inactive (MB)** | **Cost.Total (MB)** | **Vicuna** | **MT-1** | **MT-2** | **MT-Avg** | **Avg.** |
> |-----------------------------|---------------------|------------------|-----------------|------------|----------|----------|------------|----------|
> | **Setting 1: Full Participation**                                                                                                  |                   |                  |                 |            |          |          |            |          |
> | FFT-based FL                 | 13476 × 20          | 0                | 269520           | - | - | - | - | - |
> | FedAvg                       | 8.388 × 20          | 0                | 167.76           | 8.039      | 4.833    | 2.458    | 3.645      | 4.743    |
> | **FRLoRA (Ours)**                   | **8.388 × 20**      | **0**            | **167.76**       | **8.125**  | **4.984** | **2.806** | **3.895**  | **4.952** |
> | **Setting 2: Partial Participation**                                                                                              |                   |                  |                 |            |          |          |            |          |
> | FFT-based FL                 | 13476 × 2           | 0                | 26952           | - | - | - | - | - |
> | FedAvg                       | 8.388 × 2           | 0                | 16.77            | 7.925      | 4.650    | 2.025    | 3.346      | 4.486    |
> | **FRLoRA (Ours)**                   | **8.388 × 2**       | **4.194 × 18**   | **92.26**        | **8.044**  | **4.775** | **2.481** | **3.635**  | **4.733** |
>
>
> (1) In *Setting 1*,  where all clients are involved in training every round, FRLoRA incurs **NO** additional overhead, i.e., FRLoRA (167.76 MB) vs. FedAvg (167.76 MB).
>
> (2) In *Setting 2*, where only a subset of clients participates in training every round, FRLoRA adds only 75.49MB of total overhead (***Cost.Inactive***), all of which stems from parameter synchronization for inactive clients. ***Importantly***, FRLoRA indeed does **NOT** increase the communication cost per client (Only 4.194 MB for inactive clients). As the communication channels between each client and the server are independent and parallel, Cost.Inactive does **NOT** affect overall communication efficiency.
>
> (3) Besides, we further present the performance in *Setting 1*, where FRLoRA still achieves ***better*** performance than FedAvg, indicating that the performance improvement in *Setting 2* is **NOT** attributable to Cost.Inactive.
>
> **References**
>
> [1] Ye, Rui, et al. "FedLLM-Bench: Realistic Benchmarks for Federated Learning of Large Language Models." arXiv preprint arXiv:2406.04845 (2024).

---

> > ### Author Response · Authors · 2024-11-25
> > **Look forward to your post-rebuttal feedback**
> >
> > As the discussion period draws to a close soon, we extend our sincere gratitude to you for the valuable time and insightful comments.
> >
> > In our previous response, we have carefully studied your comments and made detailed responses summarized below:
> >
> > 1. Clarified used datasets include real-world use cases.
> > 2. Provided a detailed analysis of communication costs to comprehensively communication efficiency of our method.
> >
> > We sincerely hope our responses have effectively addressed your concerns. If you have any remaining questions or require further clarification, please do not hesitate to let us know, and we would be glad to provide further explanations
> >
> > Thank you again for your efforts in reviewing our work.

---

> > > ### Author Response · Authors · 2024-11-29
> > >
> > > Thanks for your valuable comment. We kindly wanted to follow up to ask if our responses have satisfactorily resolved your concerns.
> > >
> > > If you have any remaining questions or require further clarification, please do not hesitate to let us know, and we would be glad to provide further explanations.
> > >
> > > Thank you again for your efforts in reviewing our work.

---

> > > > ### Author Response · Authors · 2024-11-30
> > > >
> > > > Thanks for your valuable comment. As the discussion period draws to a close soon, we kindly wanted to follow up to ask if our responses have satisfactorily resolved your concerns.
> > > >
> > > > If you have any remaining questions or require further clarification, please do not hesitate to let us know, and we would be glad to provide further explanations.
> > > >
> > > > Thank you sincerely for your time and effort in reviewing our work.

---

### Author Response · Authors · 2024-11-23
**Official Comment by Authors**

Dear reviewers and meta-reviewers,

We appreciate all reviewers for their valuable comments and suggestions. We've revised our manuscript based on reviewers' comments as follows:

1. **For Reviewer RHQ7**, we have revised the **analysis of communication costs**, which can now be found in **Appendix C.3, Table 14** due to page limitations.

2. **For Reviewer vtoo**, we have included an analysis of **SVD computation** in **Appendix C.4, Table 15**.

3. **For Reviewer Acqq**, we have added a **clarification** in **Lines 1073-1074**.

4. **For Reviewer Acqq**, we have conducted an analysis of **client drift by cosine similarity** in **Appendix C.5, Figure 3**.

5. **For Reviewer Acqq**, we have explored the **impact of data heterogeneity** on our method for NLU tasks in **Appendix C.6, Figure 4**.

6. **For Reviewer Acqq**, we have added the results of **FlexLoRA** in **Tables 1, 2, 3, and 4**.

7. **For Reviewers Acqq and Ddf1**, we have added an analysis of the **impact of FRLoRA's initialization** in **Appendix C.7, Figure 5**.

The changes have been highlighted in **blue** in the revised paper. Please see below for our responses to each reviewer. If you have any further questions or suggestions, please feel free to share them on OpenReview.

---

### Public Comment · ~Kevin_Kuo1 · 2025-08-01
**Cites a plagiarized work**

I would like to bring to the authors' attention that this work cites an arXiv preprint (https://arxiv.org/abs/2407.20557) which has been removed due to plagiarism. The original work can be found on Google Scholar: https://scholar.google.com/scholar?cluster=577696854221514582&hl=en&as_sdt=0,47. Although ICLR 2025 has already passed, I am writing in the hope that the authors can request an update to the camera-ready and that the program committee allows for a revision.

---

### Meta-Review · Area_Chair_pGPf · 2024-12-26

**Metareview:**

This paper introduces FRLoRA, a novel approach designed to overcome the challenges of applying LoRA in a Federated Learning (FL) environment for acquiring global knowledge. Specifically, FRLoRA tackles the issues of "extrinsic client drift" and "intrinsic constrained update space."  Extensive experiments across nine benchmarks demonstrate FRLoRA's effectiveness in achieving strong performance while minimizing communication costs. To further enhance the paper, please include the reviewers' suggestions especially detailed analysis of the proposed method in the revised version.

**Additional Comments On Reviewer Discussion:**

Reviewers have requested a more in-depth analysis of the proposed algorithm including communication cost, SVD computation cost, impact over client drift, impact of FRLoRA's initialization, etc. The authors have addressed these points by providing additional experimental results and offering more detailed explanations within the revised manuscript.

---

### Decision · Program_Chairs · 2025-01-22

Accept (Poster)